# MPS1 promotes timely spindle bipolarization to prevent kinetochore-microtubule attachment errors in oocytes

Shuhei Yoshida [ID][1✉], Reiko Nakagawa [ID][2], Kohei Asai [ID][1,3] & Tomoya S Kitajima [ID][1,3✉]

## Abstract

**Incorrect kinetochore–microtubule attachment leads to chromosome segregation errors. The risk of incorrect attachment is high in acentrosomal oocytes, where kinetochores are surrounded by randomly oriented microtubules until spindle bipolarization. Regulation of the temporal relationship between acentrosomal spindle bipolarization and kinetochore–microtubule attachment is unknown. Here, we show that in mouse oocytes, MPS1, a kinase more active at kinetochores with less stable microtubule attachment, promotes timely spindle bipolarization before kinetochores stably attach to microtubules. In MPS1-inhibited oocytes, spindle bipolarization is delayed and depends on microtubules stably attached to kinetochores, resulting in incorrect attachments. We propose a two-step kinetochore-based model where unstable and stable attachment states act sequentially for acentrosomal spindle assembly to reduce the risk of egg aneuploidy.**

**Keywords** Kinetochore; Microtubule; Oocyte; Spindle
**Subject Categories** Cell Adhesion, Polarity & Cytoskeleton; Cell Cycle

## Introduction

The spindle is a microtubule-based machine that drives chromosome segregation. In somatic cells, the bipolarity of the spindle is defined by the two centrosomes that act as major microtubule-organizing centers. In contrast, in mammalian oocytes, which do not have canonical centrosomes, microtubules self-assemble a bipolar spindle with no predefined bipolar cue (Bennabi et al, 2016; Mogessie et al, 2018). Spindle assembly starts with microtubule polymerization mainly from cytoplasmic acentriolar microtubule-organizing centers in mouse oocytes (Schuh and Ellenberg, 2007) and from kinetochores in human oocytes (Holubcová et al, 2015; Wu et al, 2022), both depending on RanGTP activity elevated around chromosomes (Schuh and Ellenberg, 2007; Holubcová et al, 2015; Dumont et al, 2007). Microtubules initially form an apolar

ball-shaped spindle, which then transforms into an elongated barrel-shaped spindle through a process called spindle bipolarization. Spindle bipolarization requires the antiparallel microtubule motor KIF11 and is facilitated by numerous microtubule regulators, such as HURP, NuMA, and KIFC1/HSET (Breuer et al, 2010; Kolano et al, 2012; Bennabi et al, 2018; So et al, 2022). Many of these factors are positively regulated by the CDK1 activity (Gehmlich et al, 2004; Cahu et al, 2008; Davydenko et al, 2013), which gradually elevates through prometaphase and metaphase in oocytes (Choi et al, 1991; Davydenko et al, 2013). The CDK1 activity also promotes the stability of kinetochore–microtubule attachment (Davydenko et al, 2013), allowing its gradual increase from late prometaphase to the end of metaphase (Davydenko et al, 2013; Kitajima et al, 2011; Yoshida et al, 2015). Thus, the CDK1 activity serves as a master timer for the simultaneous progression of spindle bipolarization and kinetochore–microtubule attachment stabilization. However, whether and how the temporal relationship between these two processes is regulated remains unknown.

The temporal relationship between spindle bipolarization and kinetochore–microtubule attachment is critical for the fidelity of chromosome segregation. Kinetochores can attach to microtubules prior to spindle bipolarization, but most of such early attempts fail to properly attach the kinetochore pair of the chromosome to the opposite poles of the future bipolar spindle. These early attachments are unstable and undergo error correction after spindle bipolarization, which works efficiently and ensures faithful chromosome segregation in normal somatic cells (Foley and Kapoor, 2013). In oocytes, however, early kinetochore–microtubule attachments prior to spindle bipolarization have been implicated as a prevalent contributor to chromosome segregation errors for the following reasons (Bennabi et al, 2016; Mogessie et al, 2018; Kitajima, 2018; Mihajlović and FitzHarris, 2018). First, due to the acentrosomal nature of oocytes, microtubules are randomly oriented around kinetochores prior to spindle bipolarization, increasing the likelihood of erroneous initial attachment (Schuh and Ellenberg, 2007; Kitajima et al, 2011; Holubcová et al, 2015). Second, error correction of attachments is likely less efficient in oocytes compared to somatic cells due to the predominant regulation of attachment stabilization by CDK1 activity (Davydenko et al, 2013), which lacks specificity for correct attachments (Yoshida et al, 2015). Third, the spindle checkpoint, a mechanism

[1]Laboratory for Chromosome Segregation, RIKEN Center for Biosystems Dynamics Research (BDR), Kobe, Japan. [2]Laboratory for Cell-Free Protein Synthesis, RIKEN Center for Biosystems Dynamics Research (BDR), Kobe, Japan. [3]Graduate School of Biostudies, Kyoto University, Kyoto, Japan. ✉E-mail: shuhei.yoshida@riken.jp; tomoya.kitajima@riken.jp

that prevents anaphase entry until correct attachments are established, is less stringent in oocytes due to their large cytoplasmic size (Hoffmann et al, 2011; Kyogoku and Kitajima, 2017; Lane and Jones, 2017). Consistent with these notions, in human oocytes, a delay or instability of spindle bipolarization correlates with a subsequent increase in incorrect kinetochore–microtubule attachments (Holubcová et al, 2015). Accordingly, artificial acceleration of spindle bipolarization may reduce chromosome segregation errors (So et al, 2022). Although prioritizing spindle bipolarization over kinetochore–microtubule attachment would help prevent attachment errors, it remains unknown whether oocytes possess such a mechanism.

Recent reports provide evidence for a functional link between kinetochores and spindle bipolarization in oocytes. In mouse oocytes, NDC80, which forms a heterodimer with NUF2 and serves as a major microtubule anchor for attachment to kinetochores (Cheeseman et al, 2006; DeLuca et al, 2006), is essential for spindle bipolarization during meiosis I (Yoshida et al, 2020). NDC80-NUF2 recruits the antiparallel microtubule cross-linker PRC1 to kinetochores, which promotes KIF11-mediated spindle bipolarization (Yoshida et al, 2020). In human oocytes, kinetochore localization of PRC1 is not detected (Yoshida et al, 2020), but instead oocyte-specific microtubule-organizing centers localize to kinetochores (Wu et al, 2022). These observations are consistent with the idea that kinetochores provide a common platform for promoting acentrosomal spindle assembly via molecular mechanisms divergent among mammalian species.

In this study, we show that in mouse oocytes, MPS1, a kinase more active at kinetochores with less stably attached microtubules (Abrieu et al, 2001; Pachis and Kops, 2018), promotes timely spindle bipolarization on kinetochores unstably attached to microtubules. MPS1 exerts this function via NDC80-NUF2 at their C-terminal domains and PRC1. In MPS1-inhibited oocytes, spindle bipolarization is delayed and depends on kinetochores with stably attached microtubules, which results in incorrect kinetochore–microtubule attachments. We propose a two-step kinetochore-based model for spindle assembly, where kinetochores first promote spindle bipolarization with unstably attached microtubules and then stabilize microtubule attachment in the bipolarized spindle. This kinetochore-regulated temporal sequence —spindle bipolarization first, and stable attachment second— reduces the risk of chromosome segregation errors in acentrosomal oocytes.

# Results

## MPS1 activity ensures efficient spindle bipolarization

In mouse oocytes, kinetochores play a dual role in establishing microtubule attachment and in promoting acentrosomal spindle bipolarization (Yoshida et al, 2020). We hypothesized that kinetochores promote spindle bipolarization in response to their unattached state. One of the factors that act on unattached kinetochores is the MPS1 kinase (Abrieu et al, 2001; Pachis and Kops, 2018). To test whether MPS1 is involved in spindle bipolarization, we used the inhibitor reversine at a concentration of 1 μM, which specifically inhibited MPS1 activity but not Aurora kinase in mouse oocytes (Fig. EV1A,B) (Santaguida et al, 2010). We

added reversine to the oocyte culture immediately after inducing meiotic resumption. Oocytes were monitored for spindle formation with the microtubule marker EGFP-MAP4 and the chromosome marker H2B-mCherry by live confocal microscopy (Schuh and Ellenberg, 2007). We analyzed the dynamics of spindle morphology by measuring the sphericity and aspect ratio of an ellipsoid fitted to the 3D mass of microtubule signals (Fig. 1A). This analysis showed no detectable defects in the kinetics of spindle bipolarization in reversine-treated oocytes, except for premature anaphase spindle elongation (Fig. EV1C), consistent with previous studies (Hached et al, 2011; Yakoubi et al, 2017). We speculated that because MPS1 inhibition compromises the spindle checkpoint (Abrieu et al, 2001; Hached et al, 2011) and thereby accelerates the onset of anaphase spindle elongation, these effects may have masked a spindle bipolarization phenotype during prometaphase and metaphase. Therefore, we used proTAME, a drug that blocks anaphase entry (Zeng et al, 2010). ProTAME treatment did not significantly affect the kinetics of spindle bipolarization during prometaphase and metaphase (Fig. EV1D). However, MPS1 inhibition under the proTAME-treated condition significantly delayed spindle bipolarization (Fig. 1B; Movie EV1), without a detectable delay in initial microtubule nucleation (Fig. EV1E). This delay in spindle bipolarization was unlikely to be due to altered CDK1 activity, because proTAME treatment or additional reversine treatment did not significantly affect the temporal dynamics of the CDK1 activity sensor Eevee-spCDK (Sugiyama et al, 2024) during prometaphase and metaphase (Fig. EV2A–G). Furthermore, treatment of AZ3146, another inhibitor that inhibited MPS1 but not Aurora B/C at a concentration of 2 μM (Fig. EV1A,B) (Hewitt et al, 2010), also delayed spindle bipolarization in proTAME-treated oocytes (Fig. EV1F). In contrast, the addition of reversine after metaphase spindle establishment did not significantly affect its bipolar shape for the next 2 h (Fig. EV2H), suggesting that MPS1 activity is not required to maintain spindle bipolarity. These results suggest that MPS1 ensures efficient spindle bipolarization in oocytes.

## Defective spindle checkpoint does not delay spindle bipolarization

We wondered whether MPS1 promotes spindle bipolarization via the spindle assembly checkpoint. To address this possibility, we knocked down MAD2, a protein essential for the spindle assembly checkpoint (Homer et al, 2005), by TRIM-Away (Clift et al, 2017). MAD2 TRIM-Away significantly accelerated anaphase onset in proTAME-free oocytes (Fig. EV3A), indicating efficient disruption of the spindle assembly checkpoint (Homer et al, 2005). However, in contrast to MPS1 inhibition, MAD2 TRIM-Away did not significantly delay spindle bipolarization in proTAME-treated oocytes (Fig. EV3B). These results suggest that the role of MPS1 in spindle bipolarization is independent of the spindle assembly checkpoint.

## MPS1 activity promotes spindle bipolarization in the absence of stable kinetochore–microtubule attachment

To test the possibility that MPS1 is critical for spindle bipolarization particularly in the absence of stable kinetochore–microtubule attachment, we replaced endogenous NDC80 with NDC80-9D, a phospho-mimetic mutant form deficient in stabilizing

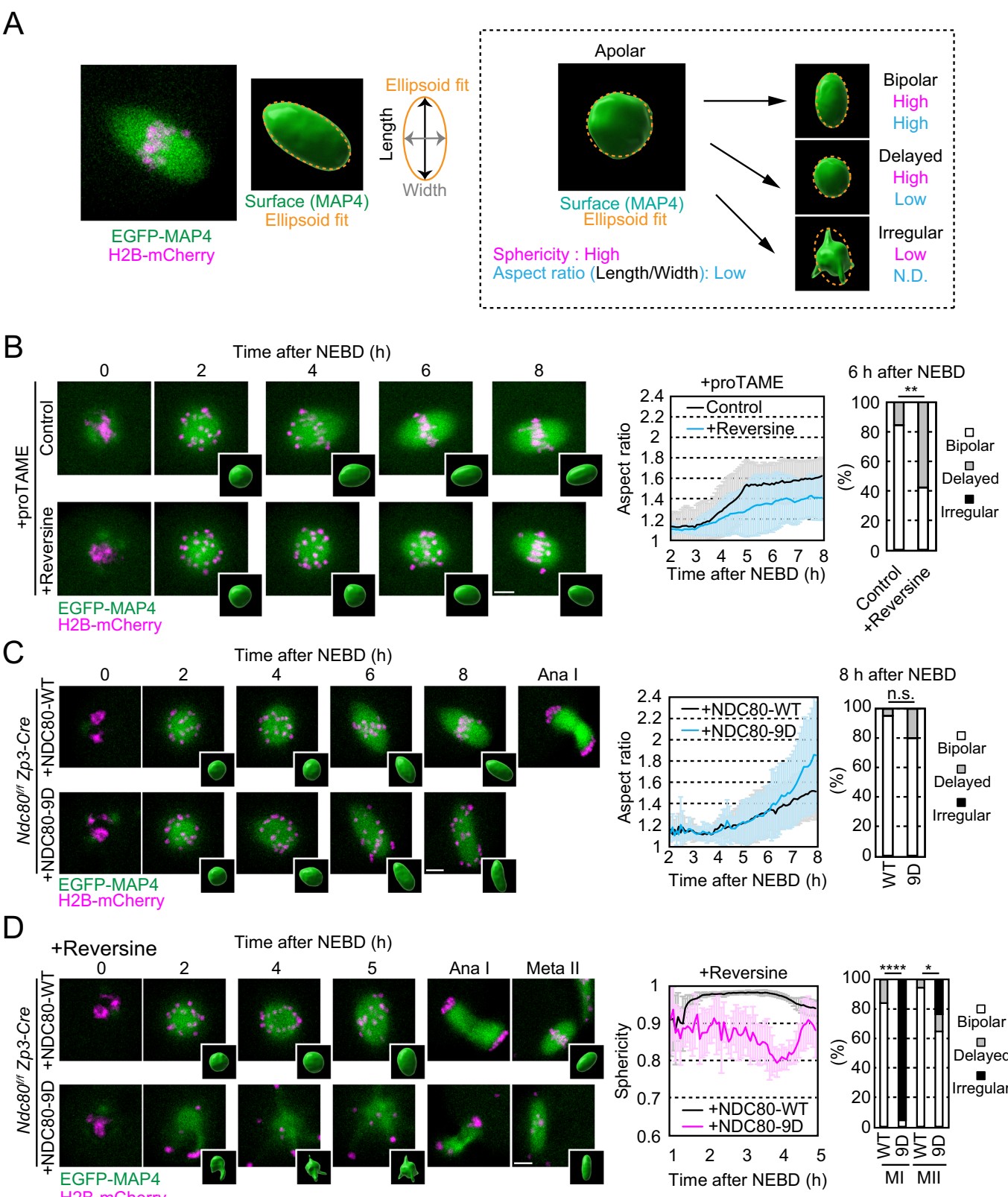

**Figure 1. MPS1 activity promotes acentrosomal spindle bipolarization in the absence of stable kinetochore–microtubule attachment.**

(A) Spindle morphology quantification. Representative z-projection image of EGFP-MAP4 (spindle, green) and H2B-mCherry (chromosome, magenta) at metaphase I in the mouse oocyte. The surface of 3D reconstructed EGFP-MAP4 is fitted to an ellipsoid. Based on the aspect ratio of the ellipsoid, spindles are classified as "bipolar" or "delayed". Spindles that did not fit well with an ellipsoid are classified as "irregular" (see "Methods"). Examples shown are identical to those in (C, D). (B) MPS1 inhibition delays spindle bipolarization. Live imaging of oocytes with proTAME and reversine. Insets show 3D reconstructed spindles. Temporal changes in the aspect ratio of the spindle (mean ± SD, $n = 14, 14$ oocytes) and morphology classification at 6 h after nuclear envelope breakdown (NEBD) ($n = 26, 26$ oocytes) are shown. Four independent experiments were performed. **$P = 0.0034$ by Fisher's exact test for "bipolar" groups. (C) Spindle bipolarization in NDC80-9D oocytes. Live imaging of $Ndc80^{f/f}$ Zp3-Cre oocytes expressing NDC80-WT/-9D. Temporal changes in the aspect ratio of the spindle (mean ± SD, $n = 6, 6$ oocytes) and morphology classification at 8 h after NEBD ($n = 21, 25$ oocytes from three independent experiments) are shown. n.s., not significant by Fisher's exact test for "bipolar" groups. (D) MPS1 inhibition impairs spindle bipolarization in NDC80-9D oocytes. Live imaging of $Ndc80^{f/f}$ Zp3-Cre oocytes expressing NDC80-WT/-9D treated with reversine. Temporal changes in the sphericity of the spindle (mean ± SD, $n = 8, 7$ oocytes) and morphology classification at 5 h after NEBD (meiosis I, MI) and MII (meiosis II) ($n = 25, 23, 18, 17$ oocytes from three independent experiments) are shown. *$P = 0.0408$, and ****$P = 0.00000001$ by Fisher's exact test for "bipolar" groups. Scale bars, 10 μm. Source data are available online for this figure.

kinetochore–microtubule attachment (Cheeseman et al, 2006; DeLuca et al, 2006; Sundin et al, 2011; Courtois et al, 2021), by deleting the floxed *Ndc80* gene with the oocyte-specific *Zp3*-Cre recombinase and exogenously expressing NDC80-9D through microinjection into oocytes (Yoshida et al, 2020; Courtois et al, 2021). Quantitative analysis showed that NDC80-9D-expressing oocytes underwent spindle bipolarization with a kinetics similar to wild-type NDC80 (NDC80-WT)-expressing oocytes, although it resulted in an excessively elongated bipolar spindle (Fig. 1C), confirming our previous report (Courtois et al, 2021). We then treated NDC80-9D-expressing oocytes with reversine. Interestingly, we found that MPS1-inhibited, NDC80-9D-expressing oocytes failed to bipolarize the spindle and exhibited an irregularly shaped spindle throughout meiosis I (Fig. 1D; Movie EV2), suggesting severe spindle bipolarization defects. These oocytes appeared to normally increase nucleated microtubules until 1 h after nuclear envelope breakdown (NEBD) but failed to accumulate them to full levels by 2 h after NEBD (Fig. EV4A), suggesting their defects in microtubule maintenance. AZ3146-treated NDC80-9D-expressing oocytes consistently showed severe defects in spindle bipolarization (Fig. EV4B). MPS1-inhibited NDC80-9D-expressing oocytes, but not NDC80-9A-expressing oocytes where kinetochore–microtubule attachments are hyperstabilized (Cheeseman et al, 2006; DeLuca et al, 2006; Sundin et al, 2011; Courtois et al, 2021), failed to form a bipolar spindle even when arrested for a prolonged period at metaphase I with proTAME (Fig. EV4C,D). In contrast, during meiosis II, MPS1 inhibition only modestly prevented spindle bipolarization in NDC80-9D-expressing oocytes (Fig. 1D; Movie EV2), consistent with previous reports that kinetochore-independent pathways support spindle bipolarization during meiosis II (Heald et al, 1996; Yoshida et al, 2020). Importantly, the expression levels of NDC80 were not affected by 9D or 9A mutation (Fig. EV4E). These observations suggest that MPS1 is required for microtubule maintenance and spindle bipolarization in the absence of stable kinetochore–microtubule attachment during meiosis I in oocytes.

## MPS1 promotes spindle bipolarization via the C-terminal domains of NDC80-NUF2

We investigated how MPS1 activity promotes spindle bipolarization. We noticed that MPS1 inhibition reduced NDC80-9D levels at kinetochores to ~73% at early prometaphase (Fig. EV5A), suggesting that MPS1 promotes kinetochore localization of

phosphorylated NDC80. Consistently, kinetochore NDC80 levels just after M-phase entry (1 h after NEBD) were significantly reduced by MPS1 inhibition, especially in oocytes treated with nocodazole, a microtubule depolymerizing drug (Fig. EV5B). Similarly, kinetochore NUF2 levels were significantly reduced by MPS1 inhibition just after M-phase entry (Fig. EV5C). However, the reduced localization of NDC80-9D in MPS1-inhibited oocytes was unlikely to sufficiently explain their severe spindle defects, because lower expression of NDC80-9D, which resulted in its kinetochore levels comparable to those of MPS1-inhibited NDC80-9D (~56%, Fig. EV5D), did not recapitulate the severe spindle defects (Fig. EV5E). These results suggest that MPS1 kinase activity promotes spindle bipolarization independently of facilitating NDC80 kinetochore localization.

We next focused on the C-terminal domains of NDC80-NUF2, which are not directly involved in microtubule attachment but promote spindle bipolarization (Yoshida et al, 2020). To specifically test whether the function of the C-terminal domains of NDC80-NUF2 in spindle bipolarization depends on MPS1 activity, we used the C-terminal fragments of NDC80 (a.a. 461–642, termed NDC80ΔN) and NUF2 (a.a. 276–463, termed NUF2ΔN) (Fig. 2A), which localize to kinetochores and partially rescue spindle bipolarization when co-expressed in *Ndc80*-deleted oocytes (Yoshida et al, 2020). Notably, MPS1 inhibition largely diminished the kinetochore localization of NDC80ΔN (Fig. EV5F), consistent with the idea that MPS1 contributes to NDC80 localization at kinetochores, and severely perturbed spindle bipolarization (Fig. 2B).

To test additional contributions of MPS1 to spindle bipolarization via the C-terminal domains of NDC80-NUF2, we tethered NDC80ΔN and NUF2ΔN to kinetochores by fusing them with the kinetochore-targeting domains of SPC25 and SPC24 (NDC80ΔN-SPC25C and NUF2ΔN-SPC24C), respectively (Yoshida et al, 2020). As expected, a substantial amount of NDC80ΔN-SPC25C (~84%) was retained at kinetochores after MPS1 inhibition (Fig. 2C). Nevertheless, MPS1 inhibition severely impaired the ability of NDC80ΔN-SPC25C and NUF2ΔN-SPC24C to rescue spindle bipolarization, resulting in the formation of an irregularly shaped spindle throughout meiosis I (Fig. 2D). This phenotype was associated with the reduced ability of NDC80ΔN-SPC25C and NUF2ΔN-SPC24C to maintain nucleated microtubules (Fig. EV5G). These results suggest that MPS1 contributes to microtubule maintenance and spindle bipolarization via the C-terminal domains of NDC80-NUF2 at kinetochores, in addition to ensuring NDC80-NUF2 localization.

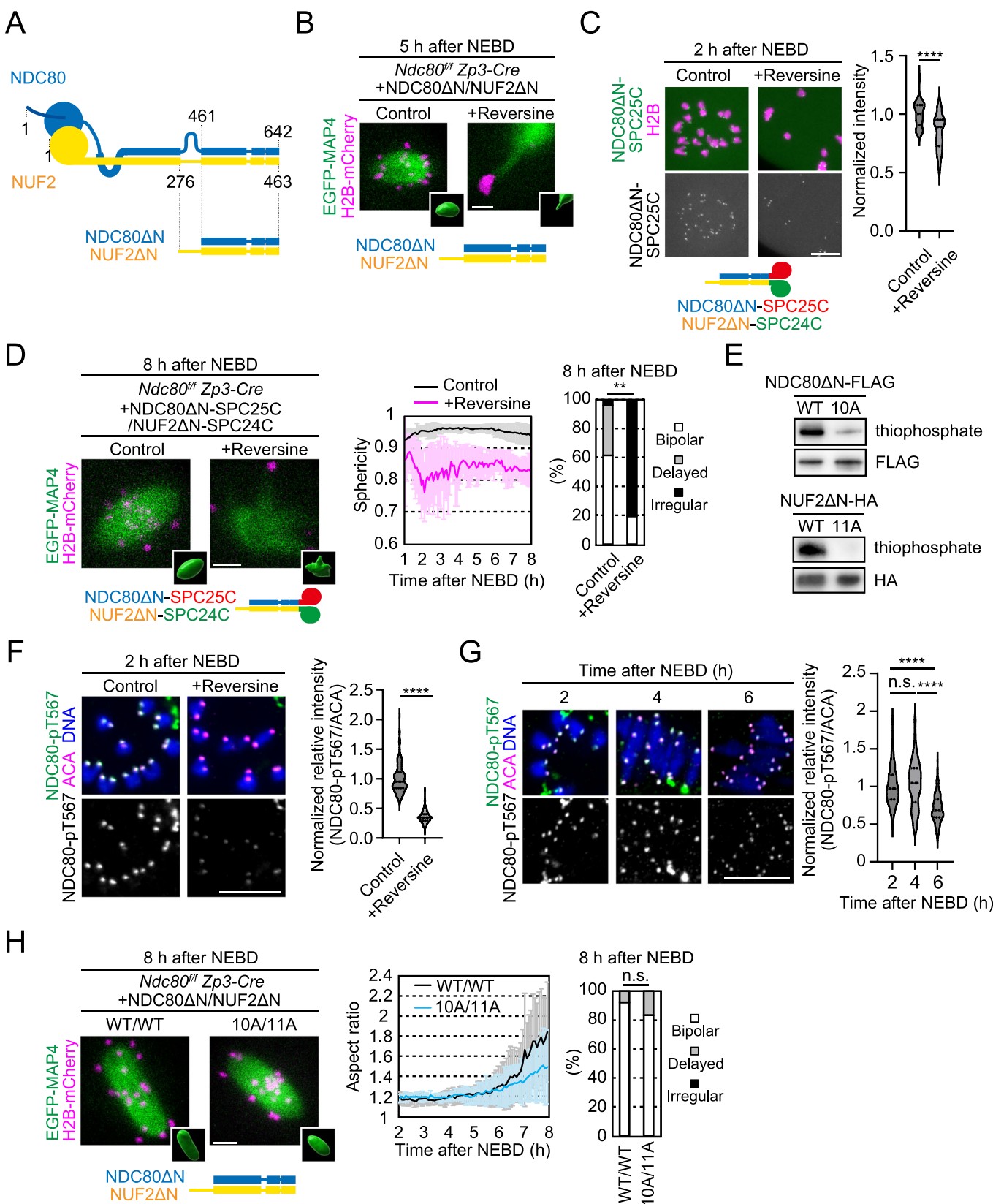

**Figure 2.  MPS1 promotes spindle bipolarization via the C-terminal domains of NDC80-NUF2 during prometaphase.**

(A) Diagram of NDC80 and NUF2. (B) The C-terminal domains of NDC80 and NUF2 require MPS1 for spindle bipolarization. Live imaging of *Ndc80^{f/f} Zp3-Cre* oocytes expressing EGFP-MAP4 (spindle, green), H2B-mCherry (chromosome, magenta), NDC80ΔN, and NUF2ΔN, treated with reversine. Z-projection and 3D-reconstruction images are shown. Three independent experiments were performed. (C) Tethering the C-terminal domains of NDC80 and NUF2 at kinetochores. Live imaging of *Ndc80^{f/f} Zp3-Cre* oocytes expressing NDC80ΔN-SPC25C-mNeonGreen, NUF2ΔN-SPC24C, and H2B-mCherry, treated with reversine. SPC25C (a.a. 120–226) and SPC24C (a.a. 122–201) are kinetochore-targeting domains. Normalized intensities of NDC80ΔN-SPC25C-mNeonGreen are shown (median and quartiles, $n = 30$, 30 kinetochores of 3, 3 oocytes. Three independent experiments were performed). ****$P = 0.000062$ by two-tailed unpaired Mann–Whitney test. (D) The C-terminal domains of NDC80 and NUF2 tethered at kinetochores require MPS1 for spindle bipolarization. Live imaging of *Ndc80^{f/f} Zp3-Cre* oocytes expressing EGFP-MAP4 (spindle, green), H2B-mCherry (chromosome, magenta), NDC80ΔN-SPC25C and NUF2ΔN-SPC24C, treated with reversine, in the presence of proTAME. Temporal changes in the sphericity of the spindle (mean ± SD, $n = 16$, 16 oocytes) and morphology classification at 8 h after NEBD ($n = 26$, 26 oocytes) are shown. Three independent experiments were performed. **$P = 0.0042$ by Fisher's exact test for "bipolar" groups. (E) Phosphorylation of NDC80ΔN and NUF2ΔN by MPS1 in vitro. Western blotting of NDC80-WT/-10A-FLAG and NUF2-WT/-11A-HA. In vitro phosphorylation was performed with ATPγS, which was detected by Western blotting against thiophosphate. (F) MPS1-dependent NDC80 phosphorylation. Immunostaining with anti-phosphorylated NDC80 at T567 (NDC80-pT567), ACA (kinetochores), and Hoechst33342 (DNA). Normalized relative intensities of NDC80-pT567 are shown (median and quartiles, $n = 200$, 200 kinetochores from 5, 5 oocytes. Three independent experiments were performed). ****$P < 0.0000000001$ by two-tailed unpaired Mann–Whitney test. (G) NDC80-T567 phosphorylation during meiosis I. Immunostaining or oocytes with anti-NDC80-pT567, ACA (kinetochores), and Hoechst33342 (DNA). Normalized relative intensities of NDC80-pT567 are shown (median and quartiles, $n = 200$, 199, 200 kinetochores from 5, 5, 5 oocytes. Three independent experiments were performed). n.s. not significant, ****$P = 0.0000000005$ by Tukey's multiple comparison test. (H) Phospho-mutants of NDC80ΔN and NUF2ΔN do not recapitulate MPS1 inhibition. Live imaging of *Ndc80^{f/f} Zp3-Cre* oocytes expressing EGFP-MAP4 (spindle, green), H2B-mCherry (chromosome, magenta), NDC80ΔN-WT/-10A and NUF2ΔN-WT/-11A. Temporal changes in the aspect ratio of the 3D reconstructed spindle (mean ± SD, $n = 10$, 12 oocytes) and morphology classification at 8 h after NEBD are shown ($n = 28$, 26 oocytes from three independent experiments). n.s., not significant by Fisher's exact test for "bipolar" groups. Scale bars, 10 μm. Source data are available online for this figure.

## MPS1 directly phosphorylates the C-terminal domains of NDC80-NUF2 during prometaphase

We searched for MPS1-mediated phosphorylation sites on the C-terminal domains of NDC80 and NUF2. In vitro kinase assay using recombinant NDC80ΔN, NUF2ΔN and MPS1 followed by mass spectrometry analysis identified 21 candidate phosphorylation sites on the C-terminal domains of NDC80 and NUF2 (NDC80-T485, T491, T494, S496, T499, S544, T567, T572, S595, and S616; and NUF2-S311, S312, S329, T336, S340, T354, T392, S403, S419, S445, and T452) (Fig. 2E). We produced phospho-specific antibodies against phosphorylated NDC80-T567, which detected kinetochores in oocytes in a manner dependent on MPS1 activity and NDC80 (Figs. 2F and EV6A). Although reversine treatment or *Ndc80* deletion did not completely abolish the phospho-antibody signals at kinetochores (Figs. 2F and EV6A), substitution of NDC80-T567 with alanine substantially reduced the phospho-antibody signals at kinetochores in oocytes (Fig. EV6B), demonstrating that a substantial fraction of the phospho-antibody signals were derived from phosphorylated NDC80-T567. Levels of NDC80-T567 phospho-antibody signals at kinetochores were high during prometaphase and decreased during metaphase (Fig. 2G), consistent with the idea that MPS1 is more active at kinetochores with less stable attachments. Overexpression of BUBR1-3A, a phospho-deficient mutant form of BUBR1 that reduces the kinetochore localization of PP2A-B56 phosphatase (Yoshida et al, 2015), significantly increased the kinetochore levels of NDC80-T567 phospho-antibody signals at late metaphase (Fig. EV6C), consistent with the idea that the gradual dephosphorylation of NDC80-T567 during metaphase is mediated by PP2A-B56. These results suggest that MPS1 phosphorylates the C-terminal domains of NDC80 and NUF2 during prometaphase, when spindle bipolarization initiates. However, substitution of all 21 candidate phosphorylation sites on the C-terminal domains of NDC80 and NUF2 (NDC80ΔN-10A and NUF2ΔN-11A), which did not affect their expression levels (Fig. EV6D), did not significantly reduce their ability to bipolarize the spindle in *Ndc80*-deleted oocytes (Fig. 2H). Thus, additional MPS1 target sites on NDC80 and NUF2,

or on other proteins, likely facilitate spindle bipolarization through the C-terminal domains of NDC80-NUF2.

## MPS1 activity promotes kinetochore localization and spindle bipolarization activity of PRC1

One of the pathways downstream of the C-terminal domains of NDC80-NUF2 is the antiparallel microtubule cross-linker PRC1, which is recruited to kinetochores and promotes spindle bipolarization (Yoshida et al, 2020). Interestingly, we found that MPS1 inhibition greatly reduced the kinetochore levels of PRC1 at early prometaphase (2 h after NEBD) (Fig. 3A). This reduction was not attributable to reduced NDC80 by MPS1 inhibition, because kinetochore NDC80 levels were not significantly decreased by MPS1 inhibition at this stage (Fig. EV7A). These results suggest that MPS1 promotes the recruitment of PRC1 to kinetochores.

Consistent with the retained ability of NDC80ΔN-10A and NUF2ΔN-11A to promote spindle bipolarization (Fig. 2H), they recruited PRC1 to kinetochores in *Ndc80*-deleted oocytes, similarly to NDC80ΔN and NUF2ΔN (Fig. EV7B). We therefore speculated that PRC1 is also a direct target of MPS1, in addition to NDC80-NUF2. Mass spectrometry analysis of recombinant PRC1 phosphorylated by MPS1 in vitro detected T40, T265, S267, T327, T379, T398, T578, and S583 as candidate phosphorylation sites (Fig. 3B). Substitution of all eight candidate phosphorylation sites (PRC1-8A) did not largely affect the ability of PRC1 to localize to kinetochores (Fig. EV7C), indicating that these phosphorylation sites are not essential for kinetochore localization of PRC1. We then explored the possibility that MPS1-mediated phosphorylation on these sites regulates PRC1 activity for spindle bipolarization independently of regulating its kinetochore localization. To evaluate the ability of PRC1 to promote spindle bipolarization independently of its kinetochore localization, we used *Ndc80*-deleted oocytes, where overexpression of PRC1 rescues spindle bipolarization defects without its kinetochore localization (Yoshida et al, 2020). We found that MPS1 inhibition significantly prevented overexpressed PRC1 from rescuing spindle bipolarization defects in *Ndc80*-deleted oocytes (Fig. 3C; Movie EV3). Notably, we found that PRC1-2A,

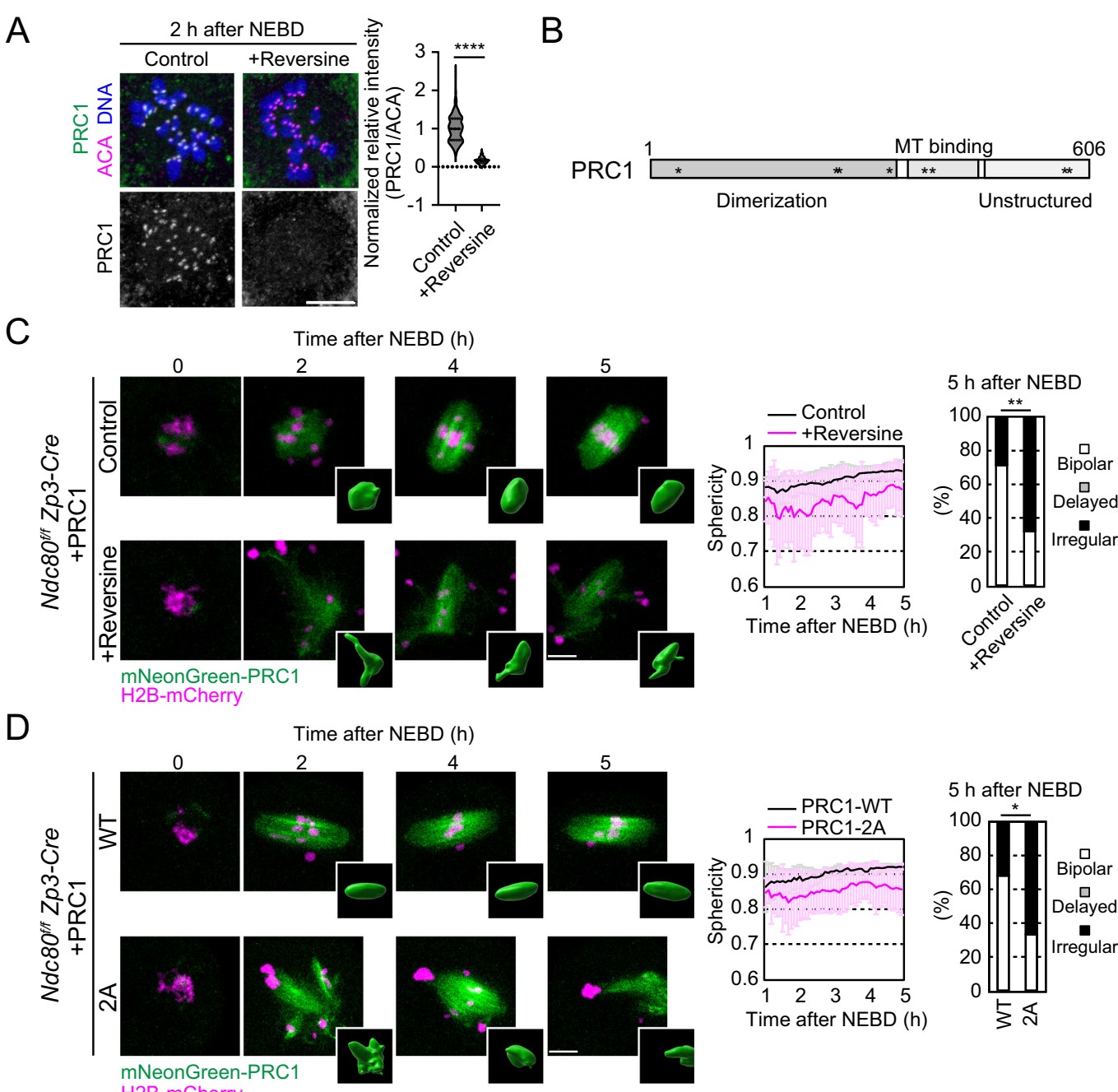

**Figure 3.  MPS1 activity promotes kinetochore localization and spindle bipolarization activity of PRC1.**

(A) MPS1 promotes kinetochore localization of PRC1. Immunostaining with anti-PRC1, ACA (kinetochores), and Hoechst33342 (DNA). Normalized relative intensities are shown (median and quartiles, $n = 240$, 240 kinetochores from 6, 6 oocytes. Three independent experiments were performed). ****$P < 0.0000000001$ by two-tailed unpaired Mann–Whitney test. (B) Phosphorylated amino acids of PRC1 by MPS1 kinase in vitro are shown. (C) PRC1 activity for spindle bipolarization depends on MPS1. Live imaging of *Ndc80^{f/f} Zp3-Cre* oocytes expressing mNeonGreen-PRC1 (green) and H2B-mCherry (chromosome, magenta), treated with reversine. Insets show 3D reconstructed images. Temporal changes in the sphericity of the spindle (mean ± SD, $n = 10$, 10 oocytes) and morphology classification at 5 h after NEBD ($n = 28$, 28 oocytes from three independent experiments) are shown. **$P = 0.0069$ by Fisher's exact test for "bipolar" groups. (D) Two potential phosphorylation sites on PRC1 are critical for its spindle bipolarization activity. As in (C), mNeonGreen-PRC1-WT/-2A (T578 and S583 substituted to alanine)-expressing *Ndc80^{f/f} Zp3-Cre* oocytes were tested (sphericity, mean ± SD, $n = 6$, 6 oocytes; morphology classification, $n = 22$, 24 oocytes from three independent experiments). *$P = 0.0377$ by Fisher's exact test for "bipolar" groups. Scale bars, 10 μm. Source data are available online for this figure.

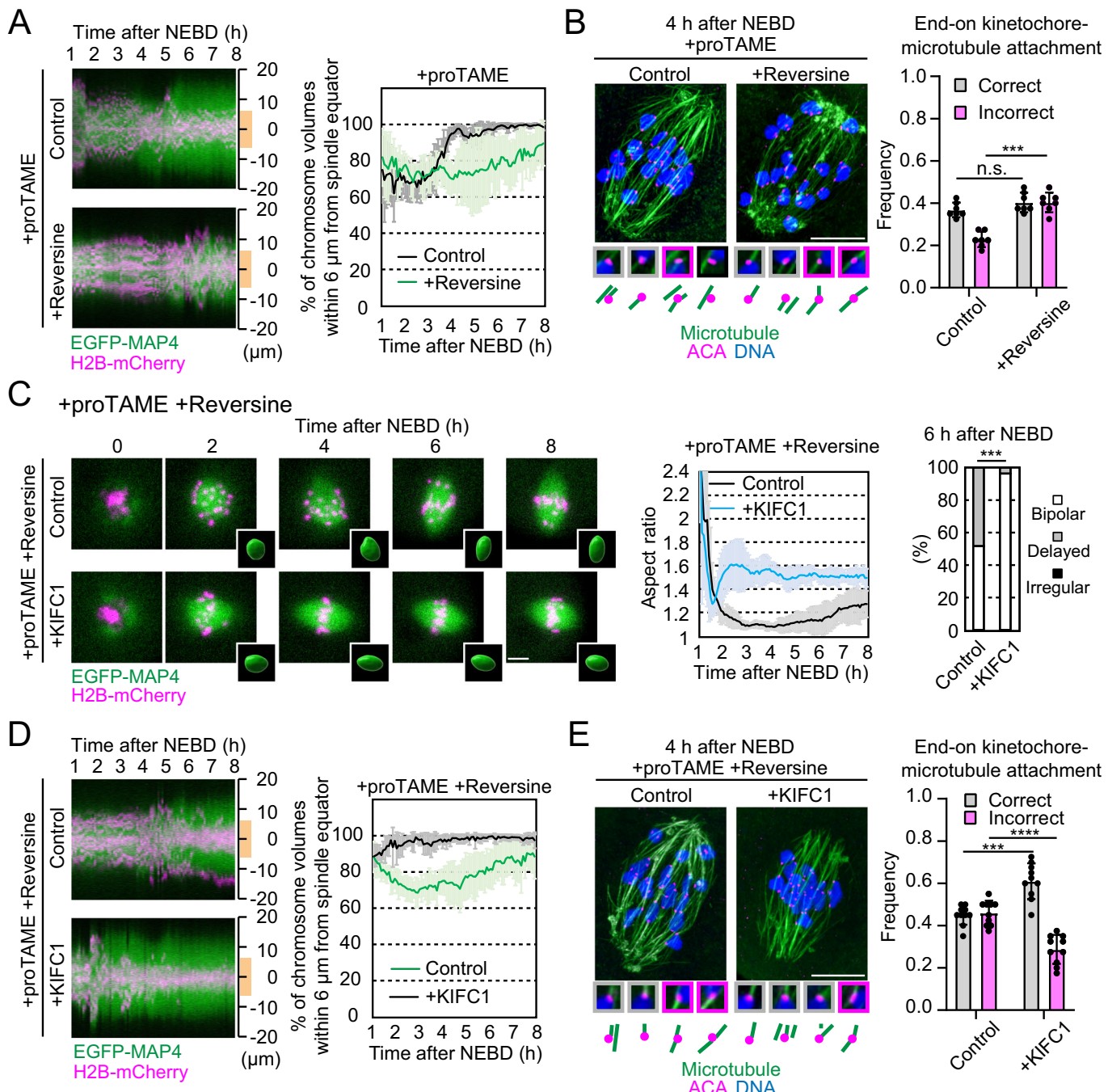

which carries alanine substitutions at two of the candidate phosphorylation sites (T578 and S583) in the C-terminal unstructured domain, largely recapitulated the failure of rescue when expressed at a level comparable to wild-type PRC1 (PRC1-WT) in *Ndc80*-deleted oocytes (Figs. 3D and EV7D; Movie EV4). PRC1-2A preferentially localized to microtubules in the middle region of the spindle (Fig. EV7E), similar to PRC1-WT, suggesting that the mutations did not affect the ability of PRC1 to crosslink antiparallel microtubules but may have affected its activity in facilitating post-crosslinking processes such as KIF11-mediated antiparallel microtubule sliding. However, phosphate affinity

polyacrylamide gel electrophoresis (Phos-tag SDS-PAGE) followed by Western blotting using prometaphase oocyte extracts did not detect MPS1-dependent band shifts of PRC1 (Fig. EV7F), providing no direct evidence for phosphorylation of these sites in vivo. Nevertheless, PRC1-2D, which carries phospho-mimetic aspartic acid substitutions at these sites, was able to rescue spindle bipolarization defects in *Ndc80*-deleted oocytes (Fig. EV7G,H), unlike PRC1-2A and similar to PRC1-WT (Fig. 3D), consistent with the idea that these sites are regulated by phosphorylation. These results suggest that MPS1 directly regulates PRC1 activity to promote spindle bipolarization.

◀ **Figure 4. MPS1-mediated timely spindle bipolarization prevents kinetochore–microtubule attachment errors.**

(A) MPS1 inhibition causes chromosome misalignment. Images of EGFP-MAP4 (spindle, green) and H2B-mCherry (chromosome, magenta) acquired in the experiments in Fig. 1B were used to generate kymographs along the spindle axis in 3D. Distance from the spindle equator is shown on the right. To analyze chromosome alignment, the percentage of chromosome volume within 6 μm from the spindle equator (orange line) was calculated (mean ± SD, $n = 7$, 7 oocytes). (B) MPS1 inhibition causes kinetochore–microtubule attachment errors. After brief cold treatment, oocytes were fixed and immunostained for stable microtubules (green), kinetochores (magenta), and DNA (blue). Magnified images of end-on monopolar (correct, gray frame) and merotelic (incorrect, magenta frame) attachments are shown (mean ± SD, $n = 7$, 7 oocytes from three independent experiments). n.s. not significant, ***$P = 0.000583$ by two-tailed unpaired Mann–Whitney test. (C) KIFC1 overexpression accelerates spindle bipolarization in MPS1-inhibited oocytes. Live imaging of oocytes expressing EGFP-MAP4 (spindle, green), H2B-mCherry (chromosome, magenta), and KIFC1, treated with reversine and proTAME. Temporal changes in the aspect ratio of the spindle (mean ± SD, $n = 9$, 9 oocytes) and morphology classification at 6 h after NEBD are shown ($n = 29$, 27 oocytes from three independent experiments). ***$P = 0.0002$ by Fisher's exact test for "bipolar" groups. (D) KIFC1 overexpression rescues chromosome misalignment in MPS1-inhibited oocytes. Oocyte images acquired in the experiment shown in (C) were analyzed for chromosome alignment as in (A) (mean ± SD, $n = 9$, 9 oocytes). (E) KIFC1 overexpression prevents kinetochore–microtubule attachment errors in MPS1-inhibited oocytes. Kinetochore–microtubule attachments in KIFC1-expressing oocytes treated with reversine and proTAME were analyzed as in (B) (mean ± SD, $n = 10$, 10 oocytes from four independent experiments). ***$P = 0.000249$, ****$P = 0.000022$ by two-tailed unpaired Mann–Whitney test. Scale bars, 10 μm. Source data are available online for this figure.

## MPS1-mediated timely spindle bipolarization prevents kinetochore–microtubule attachment errors

Our results demonstrate that MPS1 promotes spindle bipolarization via multiple pathways independent of stable kinetochore–microtubule attachment during early prometaphase. However, MPS1 is not essential for spindle bipolarization because stable attachment-dependent pathways can support spindle bipolarization in MPS1-inhibited oocytes (Fig. 1B; Movie EV1). These findings led us to ask the significance of the initiation of spindle bipolarization before stable kinetochore–microtubule attachment. In MPS1-inhibited oocytes, the delay in spindle bipolarization was accompanied by a significant increase in misaligned chromosomes (Fig. 4A; Movie EV1) and incorrect attachment of kinetochores with cold-stable microtubules at early metaphase (Fig. 4B). Thus, MPS1 promotes spindle bipolarization and prevents kinetochores from microtubule attachment errors.

We hypothesized that in MPS1-inhibited oocytes, the delay in spindle bipolarization caused chromosome misalignment with incorrect kinetochore–microtubule attachment. If this holds true, artificial acceleration of spindle bipolarization should facilitate chromosome alignment and correct kinetochore–microtubule attachment in MPS1-inhibited oocytes. To test this idea, we used overexpression of KIFC1/HSET. KIFC1/HSET is a minus-end-directed microtubule crosslinking motor that is critical for spindle pole focusing (Goshima et al, 2005). When overexpressed in oocytes, it accelerates spindle bipolarization by promoting the sorting of microtubule-organizing centers to the spindle poles, depending on its microtubule sliding activity (Bennabi et al, 2018; So et al, 2022). As expected, KIFC1 overexpression significantly accelerated spindle bipolarization in MPS1-inhibited oocytes (Fig. 4C; Movie EV5). Remarkably, these oocytes exhibited significantly rescued chromosome alignment (Fig. 4D; Movie EV5) with a significant decrease in incorrect attachment and an increase in correct attachment of kinetochores with cold-stable microtubules (Fig. 4E). These results suggest that MPS1 activity prevents incorrect kinetochore–microtubule attachment by timely initiating spindle bipolarization during early prometaphase.

## MPS1 is not required for aligning NDC80-NUF2-tethered microbeads or suppressing post-metaphase incorrect kinetochore–microtubule attachment

We considered the possibility that MPS1 prevents chromosome misalignment by directly regulating NDC80-NUF2-mediated microtubule attachment. We recently reported that NDC80-NUF2-tethered microbeads efficiently align at the metaphase plate with cold-unstable microtubule attachments (Asai et al, 2024), allowing us to test the requirement of MPS1 for NDC80-NUF2-mediated alignment mechanisms. We found that MPS1 inhibition did not significantly increase the misalignment of NDC80-NUF2-tethered microbeads (Fig. EV8A,B). These results suggest that MPS1 is dispensable for NDC80-NUF2-mediated mechanisms that promote alignment with cold-unstable microtubule attachments.

We also tested the possibility that MPS1 functions after spindle bipolarization to prevent incorrect kinetochore–microtubule attachment. Reversine treatment after metaphase spindle establishment did not significantly increase incorrect attachments of kinetochores with cold-stable microtubules nor decrease correct attachments (Fig. EV8C). These results suggest that MPS1 activity after spindle bipolarization is dispensable for suppressing incorrect kinetochore–microtubule attachment.

## Discussion

In oocytes, due to the absence of centrosomes, kinetochores are surrounded by randomly oriented microtubules during early prometaphase prior to spindle bipolarization. This situation favors kinetochores to form improper microtubule attachments, which can lead to chromosome segregation errors (Bennabi et al, 2016; Kitajima, 2018; Mihajlović and FitzHarris, 2018; Mogessie et al, 2018). In this study, we identified the MPS1 kinase as a key player that allows kinetochores to actively promote spindle bipolarization before stabilizing their microtubule attachments. MPS1 kinase activity is essential for spindle bipolarization when kinetochore–microtubule attachment is unstable, by regulating multiple downstream factors, including the C-terminal regions of NDC80 and NUF2, and PRC1. Oocytes lacking MPS1 kinase activity exhibit delayed spindle bipolarization with stable kinetochore–microtubule attachments, which result in incorrect kinetochore–microtubule attachment. Thus, MPS1 prevents incorrect kinetochore–microtubule attachment by promoting timely spindle bipolarization at kinetochores.

Based on these findings, we propose a two-step model where kinetochores act sequentially with unstable and stable microtubule attachments for acentrosomal spindle assembly (Fig. 5). In the first mode, unstably attached kinetochores employ MPS1 activity to create a microenvironment that concentrates microtubule regulators, such as the antiparallel microtubule cross-linker PRC1, via NDC80-NUF2

(Yoshida et al, 2020). This microenvironment facilitates microtubule maintenance and KIF11-mediated bipolar microtubule sorting, thereby initiating spindle bipolarization during early prometaphase. The acquisition of spindle bipolarity increases the likelihood that microtubules from both poles properly attach to the kinetochore pair of the chromosome. Kinetochores drive the first mode until the gradual increase in CDK1 activity stabilizes microtubule attachment (Davydenko et al, 2013; Yoshida et al, 2015), which reduces MPS1 activity at kinetochores (Ji et al, 2015; Hiruma et al, 2015) and thereby allows kinetochores to switch to the second mode. In the second mode, kinetochores employ stably attached microtubules, which complete spindle bipolarization while preventing excessive elongation during late prometaphase and metaphase (Courtois et al, 2021). In oocytes lacking MPS1 activity, the first mode is absent and thus spindle bipolarization is delayed, while increased CDK1 activity stabilizes microtubules attached to kinetochores from random directions. The microtubules stably attached to kinetochores initiate the second mode of spindle bipolarization and thus remain as incorrect, merotelic attachments in the resulting bipolar spindle. According to this model, in normal oocytes, kinetochores self-regulate the temporally sequential action of the two distinct modes for kinetochore-based bipolar spindle assembly. This kinetochore-regulated temporal sequence—spindle bipolarization first, and stable attachment second—prevents the formation of incorrect kinetochore–microtubule attachments (Davydenko et al, 2013).

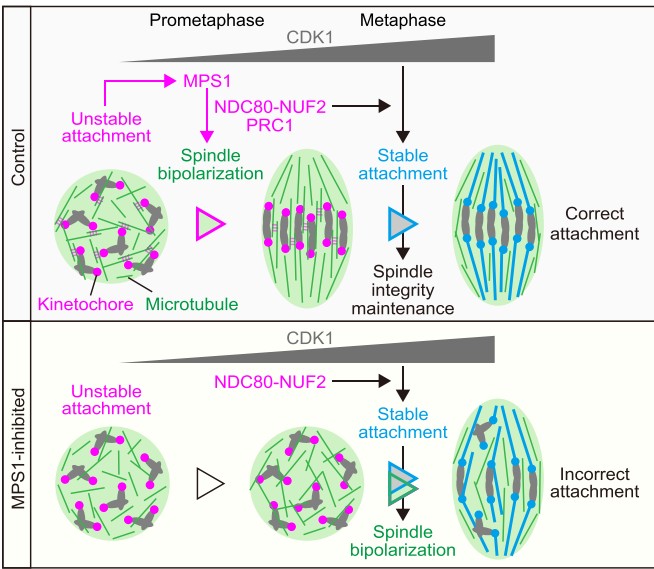

**Figure 5. A two-step kinetochore-based model for acentrosomal spindle assembly that prevents microtubule attachment errors in oocytes.**

During prometaphase, kinetochores with unstable microtubule attachment employ MPS1 to promote microtubule maintenance and spindle bipolarization via the C-terminal regions of NDC80-NUF2, and the antiparallel microtubule cross-linker PRC1. Spindle bipolarity facilitates bipolar microtubule attachment to the kinetochore pair of chromosomes. Subsequently, a gradual increase in CDK1 activity stabilizes kinetochore–microtubule attachment. The stably attached microtubules complete spindle bipolarization. However, in oocytes lacking MPS1 activity, spindle bipolarization is delayed, while CDK1 activity gradually increases, which stabilizes kinetochore–microtubule attachment. The microtubules stably attached to kinetochores participate in spindle bipolarization and thus remain as merotelic microtubules in the bipolar spindle.

Our results show that one of the critical roles of MPS1 kinase in oocytes is the timely initiation of spindle bipolarization, in addition to spindle checkpoint activation and centromeric cohesion protection (Hached et al, 2011; Yakoubi et al, 2017). In somatic cells, chromosome misalignment induced by MPS1 inhibition is attributed to loss of MPS1-mediated direct regulation of kinetochore–microtubule attachment (Jelluma et al, 2008; Yamagishi et al, 2012; Maciejowski et al, 2017). In contrast, in oocytes, a substantial fraction of misaligned chromosomes induced by MPS1 inhibition are likely attributed to delayed spindle bipolarization, as chromosome misalignment and incorrect kinetochore–microtubule attachment in MPS1-inhibited oocytes are largely suppressed by overexpression of KIFC1, which artificially accelerated spindle bipolarization. Although we cannot formally exclude the possibility that KIFC1 directly promotes chromosome alignment, based on the activity of KIFC1 to directly promote microtubule crosslinking and sliding (Goshima et al, 2005; Cai et al, 2009; Bennabi et al, 2018; So et al, 2022), we suggest that KIFC1 suppressed chromosome misalignment by accelerating spindle bipolarization.

Although this study showed the requirement of MPS1 for timely spindle bipolarization during meiosis I in oocytes, it remains unclear whether MPS1 plays a similar role in meiosis II and somatic mitosis. In contrast to meiosis I, bipolar spindle assembly in meiosis II does not rely on kinetochores but is largely mediated by chromosome-dependent pathways (Yoshida et al, 2020; Heald et al, 1996), suggesting the possibility that the kinetochore MPS1-dependent pathway may be less critical for spindle bipolarization in meiosis II. In somatic mitosis with centriole-containing centrosomes, centrosomes play an important role in bipolar spindle assembly. We suggest that the MPS1-mediated timely initiation of spindle bipolarization is particularly critical for meiosis I in acentrosomal oocytes. It is still possible that the MPS1-dependent kinetochore pathway acts as a back-up mechanism for bipolar spindle assembly in meiosis II and somatic mitosis.

MPS1 kinase activity promotes spindle bipolarization likely through multiple pathways in oocytes. First, MPS1 ensures rapid and proper localization of NDC80 to kinetochores within the short time window of 1 h after NEBD. Second, MPS1 regulates the C-terminal regions of NDC80 and NUF2. Our data show that MPS1 directly phosphorylates these regions in oocytes, although it is unclear whether direct phosphorylation contributes to timely spindle bipolarization. Third, MPS1 regulates PRC1, an antiparallel microtubule cross-linker that promotes spindle bipolarization (Yoshida et al, 2020; Bieling et al, 2010; Subramanian et al, 2010). MPS1 promotes kinetochore localization and spindle bipolarizing activity of PRC1. It is likely that MPS1 recruits PRC1 via the C-terminal domains of NDC80-NUF2, as these domains, when tethered to kinetochores, rescue the kinetochore localization defects of PRC1 in *Ndc80*-deleted oocytes (Yoshida et al, 2020) in an MPS1-dependent manner (Figs. 3A and EV5F). Although a physical interaction of the C-terminal domains of NDC80-NUF2 with PRC1 is suggested by the yeast two-hybrid assay (Yoshida et al, 2020), future work should test this with purified proteins. Furthermore, the MPS1-mediated phosphorylation sites on NDC80, NUF2 or PRC1 identified in this study were not required for their kinetochore localization, suggesting that critical phosphorylation sites remain to be identified. Since MPS1 inhibition perturbs PRC1-dependent spindle bipolarization in *Ndc80*-deleted oocytes (Fig. 3C), where PRC1 does not localize to kinetochores (Yoshida et al, 2020), MPS1 likely can phosphorylate

PRC1 in the cytoplasm, although the kinetochore localization of MPS1 likely facilitates the phosphorylation of PRC1, NDC80 and NUF2 at kinetochores in normal oocytes. Importantly, our data do not exclude the possibility that MPS1 has additional downstream pathways to promote spindle bipolarization. Although future studies are needed to fully understand the downstream molecular pathways of MPS1, our current results indicate that MPS1 is a master kinase that promotes timely spindle bipolarization at kinetochores prior to stabilization of microtubule attachment.

Kinetochores contribute to bipolar spindle assembly through distinct mechanisms in mouse and human oocytes. In mouse oocytes, kinetochores with unstable microtubule attachment employ MPS1 to promote spindle bipolarization, whereas in human oocytes, kinetochores recruit microtubule-organizing centers to promote microtubule polymerization (Wu et al, 2022). Whether human oocytes require MPS1 kinase activity for the recruitment of microtubule-organizing centers, microtubule polymerization, or spindle bipolarization is an intriguing question for future studies.

# Methods

### Reagents and tools table

| Reagent/resource | Reference or source | Identifier or catalog number |
|---|---|---|
| **Experimental models** | | |
| B6D2F1/Slc | Japan SLC | (C57BL/6NCrSlc ♀×DBA/2CrSlc ♂) F1 |
| *Ndc80^flox/flox* Zp3-Cre | Yoshida et al, 2020 | |
| **Recombinant DNA** | | |
| pGEMHE_EGFP-MAP4 | Schuh and Ellenberg, 2007 | |
| pGEMHE_H2B-mCherry | Kitajima et al, 2011 | |
| pGEMHE_NDC80-WT | Courtois et al, 2021 | |
| pGEMHE_NDC80-9A | Courtois et al, 2021 | |
| pGEMHE_NDC80-9D | Courtois et al, 2021 | |
| pGEMHE_NDC80-T567A | This study | |
| pGEMHE_NDC80ΔN-WT | Yoshida et al, 2020 | |
| pGEMHE_NDC80ΔN-10A | This study | |
| pGEMHE_ NDC80ΔN-mNeonGreen | Yoshida et al, 2020 | |
| pGEMHE_NUF2ΔN-WT | Yoshida et al, 2020 | |
| pGEMHE_NUF2ΔN-11A | This study | |
| pGEMHE_NDC80ΔN-SPC25C | Yoshida et al, 2020 | |
| pGEMHE_NDC80ΔN-SPC25C-mNeonGreen | This study | |
| pGEMHE_NUF2ΔN-SPC24C | Yoshida et al, 2020 | |
| pGEMHE_mNeonGreen-PRC1-WT | This study | |
| pGEMHE_mNeonGreen-PRC1-2A | This study | |
| pGEMHE_mNeonGreen-PRC1-2D | This study | |

| Reagent/resource | Reference or source | Identifier or catalog number |
|---|---|---|
| pGEMHE_NDC80-9D-GFP | This study | |
| pGEMHE_KIFC1 | This study | |
| pGEMHE_24xGCN-PRC1-WT | Yoshida et al, 2020 | |
| pGEMHE_24xGCN-PRC1-8A | This study | |
| pGEMHE_scFvsfGFP | Ding et al, 2018 | |
| pGEMHE_ NDC80ΔN-mEGFP | This study | |
| pGEMHE_NDC80-WT-mNeonGreen | This study | |
| pGEMHE_NDC80-9A-mNeonGreen | This study | |
| pGEMHE_NDC80-9D-mNeonGreen | This study | |
| pGEMHE_mEGFP-BUBR1-WT | Yoshida et al, 2015 | |
| pGEMHE_mEGFP-BUBR1-3A | Yoshida et al, 2015 | |
| pGEMHE_TRIM21 | Clift et al, 2017 | |
| pGEMHE_Eevee-spCDK | This study | |
| pFastBac_GST- NDC80ΔN-WT-FLAG | This study | |
| pFastBac_GST- NDC80ΔN-10A-FLAG | This study | |
| pFastBac_NUF2ΔN-WT-HA | This study | |
| pFastBac_NUF2ΔN-11A-HA | This study | |
| pFastBac_GST-Prc1-His | This study | |
| **Antibodies** | | |
| Rat anti-Histone H3 (pS28) | Abcam | ab10543 |
| Rabbit anti-Histone H3 | Abdam | ab62706 |
| Rabbit anti-NDC80 | Diaz-Rodriguez et al, 2008 | |
| Human anti-centromere protein antibody | Antibodies Incorporated | 15-234 |
| Rat monoclonal anti-alpha tubulin | Bio-Rad | MCA77G |
| Rat anti-GFP antibody | Nacalai | GF090R 04404-84 |
| Rabbit anti-PRC1 | Santa Cruz | sc-8356 |
| Alexa Fluor 488 goat anti-rabbit IgG (H + L) | Thermo Fisher | A11034 |
| Alexa Fluor 488 goat anti-rat IgG (H + L) | Thermo Fisher | A11006 |
| Alexa Fluor 555 goat anti-human IgG (H + L) | Thermo Fisher | A21433 |
| Alexa Fluor 555 donkey anti-rabbit IgG (H + L) | Thermo Fisher | A31572 |
| Alexa Fluor 647 donkey anti-rabbit IgG (H + L) | Thermo Fisher | A31573 |
| Rabbit polyclonal antibody against NDC80-pT567 | This study | |
| Mouse anti-MAD2 antibody | Santa Cruz | sc-65492 |
| Mouse anti-FLAG antibody | Sigma-Aldrich | F1804 |
| Rat anti-HA antibody | Roche | 11867423001 |

| Reagent/resource | Reference or source | Identifier or catalog number |
|---|---|---|
| Rabbit anti-thiophosphate ester antibody | Abcam | ab92570 |
| Rabbit anti-PRC1 antibody | Proteintech | 15617-1-AP |
| Rabbit anti-GFP antibody | Abcam | ab6556 |
| Rabbit anti-NUF2 antibody | Abcam | ab230313 |
| Rabbit anti-NDC80 antibody | He et al, 2023 | |
| Rabbit anti-actin antibody | Abcam | ab1801 |
| Goat anti-Mouse IgG (H + L) Secondary Antibody, HRP | Thermo Fisher | 31430 |
| Goat anti-Rat IgG (H + L) Secondary Antibody, HRP conjugate | Life | A18865 |
| Goat anti-Rabbit IgG(H + L) Secondary antibody, HRP conjugate | Life | A16104 |
| **Oligonucleotides and other sequence-based reagents** | | |
| **Chemicals, enzymes, and other reagents** | | |
| CARD HyperOva | KYUDO | F-021 |
| mMESSAGE mMACHINE T7 kit | invitrogen | AM1344 |
| 3-isobutyl-1-methyl-xanthine | Sigma-Aldrich | 15879 |
| Reversine | Cayman | 10004412 |
| AZ3146 | Selleck | S2731 |
| proTAME | RD systems | 554-17621 |
| Nocodazole | Sigma-Aldrich | M1404 |
| Anti-GFP mAb-Magnetic Beads | MBL | D153-11 |
| GST SpinTrap | Cytiva | 28952359 |
| PreScission Protease | Cytiva | 27-0843-01 |
| cOmplete, EDTA-free | Roche | 11873580001 |
| TTK | Carna Biosciences | 05-169 |
| ATPγS | Abcam | ab138911 |
| PNBM | Abcam | ab138910 |
| **Software** | | |
| Fiji | https://fiji.sc/ | |
| Imaris | Oxford Instruments | |
| GraphPad Prism 9 | GraphPad Software | |
| **Other** | | |
| SuperSepTM Phos-tagTM (50µmol/L), 10%, 13well | FUJIFILM Wako Pure Chemical Corporation | 193-16711 |

## Methods and protocols

### Mice

All animal experiments were approved by the Institutional Animal Care and Use Committee of RIKEN Kobe Branch (IACUC). B6D2F1 (C57BL/6 x DBA/2), *Ndc80^{flox/flox} Zp3-Cre* female mice (Yoshida et al, 2020), 8–16 weeks old, were used to obtain oocytes.

### Mouse oocyte culture

Mice were injected with 0.1 ml of CARD HyperOva (KYUDO). Fully grown oocytes were collected 48 h after injection and placed in M2 medium containing 200 µM 3-isobutyl-1-methyl-xanthine (IBMX, Sigma) at 37 °C. Meiotic resumption was induced by washing to remove IBMX. When indicated, 1 µM reversine (Cayman), 2 µM AZ3146 (Selleck), 5 µM proTAME (RD systems), 66 µM nocodazole (Sigma) were used. DMSO was used as a control. Unless otherwise indicated, the drugs were added to the oocyte culture immediately after the induction of meiotic resumption.

### mRNA synthesis and injection

mRNAs were transcribed in vitro by using the mMESSAGE mMACHINE T7 kit (Invitrogen). The mRNAs were introduced into fully grown oocytes by microinjection. The microinjected oocytes were cultured at 37 °C for 3–4 h before IBMX washing. Microinjections were performed with mRNA of EGFP-MAP4 (3 pg); H2B-mCherry (0.15 pg); NDC80-WT, −9A, −9D, and T567A (1 pg unless otherwise indicated); NUF2-HA (1 pg); NDC80ΔN-WT and −10A (1 pg); NDC80ΔN-mNeonGreen (1 pg); NUF2ΔN-WT and −11A (1 pg); NDC80ΔN-SPC25C (1 pg); NDC80ΔN-SPC25C-mNeonGreen (1 pg); NUF2ΔN-SPC24C (1 pg); mNeonGreen-PRC1-WT, −2A and −2D (2 pg); NDC80-9D-GFP (1 pg, 0.1 pg); KIFC1 (0.12 pg); 24xGCN-PRC1-WT, −8A (0.1 pg); scFv-sfGFP (0.25 pg); NDC80ΔN-mEGFP (1 pg); NDC80-WT, −9A, and −9D-mNeonGreen (1 pg); mEGFP-BUBR1-WT, −3A (30 pg), TRIM21 (1 pg) and Eevee-spCDK (10 pg).

### MAD2 TRIM-Away

For MAD2 TRIM-Away, a mouse anti-MAD2 antibody (sc-65492, Santa Cruz, 0.1 pg) was injected into fully grown oocytes with mRNAs by microinjection. PBS was used as a control. The microinjected oocytes were cultured at 37 °C for 3–4 h before IBMX washout.

### NDC80-NUF2 microbead injection

The mRNAs of NDC80-GFP (1 pg), NUF2 (1 pg), and H2B-mCherry (1.5 pg) were introduced into fully grown oocytes by microinjection. The mRNA-microinjected oocytes were cultured at 37 °C for 2 h, and then microinjected with Anti-GFP mAb-Magnetic Beads (2.0 µm in diameter, D153-11, MBL). Three beads were microinjected into each oocyte. The oocytes were cultured at 37 °C for 1 h before IBMX washout.

### Live imaging

A customized Zeiss LSM710, LSM780, or LSM880 confocal microscope equipped with a 40x C-Apochromat 1.2 NA water immersion objective lens (Carl Zeiss) was controlled by Zen software using the multi-position autofocus macros AutofocusScreen (Rabut and Ellenberg, 2004) and MyPic (Politi et al, 2018). For spindle and chromosome imaging, we recorded 11 z-confocal sections (every 4 µm) of 512 × 512 pixel xy images at 5–6 min time intervals for at least 12 h after the induction of meiotic resumption. For kinetochore imaging, we recorded 17 z-confocal sections (every 1.5 µm) of 512 × 512 pixel xy images at 5–6 min time intervals for at least 12 h after the induction of meiotic resumption. For microbead imaging, we recorded 25 z-confocal sections (every 1.25 µm) of

$512 \times 512$ pixel xy images at 5–6 min time intervals for at least 12 h after the induction of meiotic resumption.

### 4D spindle analysis

To analyze spindle morphology, we performed 3D surface rendering of EGFP-MAP4 or mNeonGreen-PRC1 signals using Imaris software (Oxford Instruments). Images covering almost the entire spindle were used for the analysis, and others were excluded. The generated 3D surface was fitted to an ellipsoid, which was used to categorize spindle morphology. If the volume of the fitted ellipsoid was >1.05-fold greater than that of the 3D surface of the spindle, indicating that the spindle did not fit well to an ellipsoid, the spindle was classified as "irregular". Otherwise, the aspect ratio of the length (longest axis length) to the width (average of two shorter axis lengths) of the fitted ellipsoid and its sphericity were calculated. Spindles with an aspect ratio greater than 1.2 were classified as "bipolar", and others as "delayed".

### Immunostaining of oocytes

Oocytes were fixed with 1.6% formaldehyde (methanol-free) in 10 mM PIPES (pH 7.0), 1 mM $MgCl_2$, and 0.1% Triton X-100 for 30 min at room temperature. To detect cold-stable microtubules, oocytes were incubated in ice-cold M2 medium for 10 min before fixation (cold treatment). After fixation, oocytes were washed and permeabilized with PBT (PBS with 0.1% Triton X-100) at 4 °C overnight. After blocking with 3% bovine serum albumin (BSA)-PBT for 1 h, oocytes were incubated with primary antibodies at 4 °C overnight. Oocytes were washed with 3% BSA-PBT and then incubated with secondary antibodies and 20 µg/ml Hoechst33342 for 2 h for kinetochore imaging or overnight for cold-stable microtubule imaging. Oocytes were washed again and stored in 0.01% BSA-PBS. Oocytes were imaged using a Zeiss LSM780, LSM980 confocal microscope for kinetochore imaging, or a Zeiss LSM880 confocal microscope with AiryScan for cold-stable microtubule imaging.

The following primary antibodies were used: a rat anti-Histone H3 (pS28) (1:500, ab10543, Abcam), a rabbit anti-Histone H3 (1:500, ab62706, Abcam), a rabbit anti-NDC80 antiserum (1:2000, a gift from Dr. Robert Benezra), a human anti-centromere protein antibody (ACA, 1:500, 15-234, Antibodies Incorporated), a rat monoclonal anti-alpha tubulin (1:2000, MCA77G, Bio-Rad), a rat anti-GFP antibody (1:500, GF090R 04404-84, Nacalai), and a rabbit anti-PRC1 (1:100, H-70 sc-8356, Santa Cruz). The following secondary antibodies were used: Alexa Fluor 488 goat anti-rabbit IgG (H + L) (A11034), goat anti-rat IgG (H + L) (A11006), Alexa Fluor 555 goat anti-human IgG (H + L) (A21433), donkey anti-rabbit IgG (H + L) (A31572), and Alexa Fluor 647 donkey anti-rabbit IgG (H + L) (A31573) (1:500, Thermo Fisher).

### Phospho-specific antibodies against phosphorylated NDC80-T567

To produce a rabbit polyclonal antibody against NDC80-pT567, the phosphopeptide Cys+QREYQL(pT)VKTTT was synthesized and used for the immunization of a rabbit. The anti-NDC80-pT567 antibody was affinity-purified from the immunized serum using the same phosphopeptide. The fraction that binds to the non-phosphorylated peptide Cys+QREYQLTVKTTT was absorbed.

### Quantification of signal intensity

Fiji (https://fiji.sc/) and Imaris software (Oxford Instruments) were used to quantify fluorescence signals. We measured fluorescence intensity for NDC80, PRC1, or GFP at kinetochores and subtracted the signal intensity at a cytoplasmic region near the kinetochore. Similarly, we measured the fluorescence intensity for ACA at the same kinetochores. We calculated the ratio of fluorescence intensity of NDC80, PRC1, or GFP to that of ACA.

### FRET

The FRET (YFP/CFP) ratio was quantified as the ratio of fluorescence intensity of YFP to CFP. Fiji (https://fiji.sc/) was used to quantify fluorescence signals. We measured fluorescence intensity for YFP in a cytoplasmic region near chromosomes. Similarly, we measured fluorescence intensity for CFP in the same cytoplasmic region. After subtracting the background (outside the oocyte), we calculated the ratio of fluorescence intensity of YFP to CFP. FRET ratio images were visualized by the "16 colors" color code.

### Protein purification

Bacmid DNA containing GST-NDC80ΔN-FLAG, NUF2ΔN-HA, or GST-PRC1-His was transfected into Sf9 insect cells to produce baculovirus. For protein expression, baculovirus-infected cells were grown at 28 °C for 48 h. After washing with PBS, the cells were stored at −80 °C. For protein purification, cells were suspended in 50 mM HEPES (pH 7.4), 300 mM NaCl, 1 mM EDTA, 10% glycerol, 1% Triton X-100, 1 mM DTT and protease inhibitor cocktail (cOmplete EDTA-free, Roche) and lysed by sonication on ice. After centrifugation at 14,000 rpm for 60 min, the soluble fraction of the cell lysate was bound to glutathione-sepharose 4B (GST SpinTrap, Cytiva). The unbound fraction was washed with 50 mM HEPES (pH 7.4), 250 mM NaCl, 1 mM EDTA, 10% glycerol and 1 mM DTT. Proteins were eluted by cleavage with PreScission protease (Cytiva), which cleaves GST in 50 mM HEPES (pH 7.4), 150 mM NaCl, 1 mM EDTA, 0.05% Triton X-100 and 1 mM DTT.

### In vitro kinase reaction

Purified proteins were incubated with 200 ng human MPS1 (TTK, 05-169, Carna Biosciences) in 16 mM KCl, 32 mM Na-β-glycerophosphate, 8 mM EGTA, 6 mM $MgCl_2$, 0.4 mM DTT, and 2.5 mM ATP at 37 °C for 1 h. To detect phosphorylation, 1 mM adenosine 5'-O-(3-thiotriphosphate) (ATPγS, ab138911, abcam) was used instead of ATP for the kinase reaction. After kinase reaction, p-nitrobenzyl mesylate (PNBM, ab138910, abcam) was added (final 2.5 mM) and incubated for 1 h at room temperature.

### Western blotting

Fully grown oocytes were microinjected with mRNAs and cultured at 37 °C for 3 h. Proteins were prepared in an SDS-PAGE sample buffer. After heating at 95 °C for 5 min, they were detected SDS-PAGE followed by Western blotting. For Fig. EV7F, Phos-tag SDS-PAGE was performed according to the manufacturer's instructions (193-16711, FUJIFILM Wako Pure Chemical Corporation). The primary antibodies were a mouse anti-FLAG antibody (F1804, Sigma-Aldrich), a rat anti-HA antibody (11867423001, Roche), a rabbit anti-thiophosphate ester antibody (ab92570, abcam) (1:2000), a rabbit anti-PRC1 antibody (15617-1-AP, proteintech), a rabbit anti-GFP antibody (ab6556, abcam), a rabbit anti-NUF2 antibody (ab230313, abcam), a rabbit anti-NDC80 antibody (gift from Dr. Hiroki Shibuya) and a rabbit anti-actin antibody (ab1801, abcam). Horseradish peroxidase-conjugated anti-mouse, anti-rat,

### Mass spectrometry (MS)

Proteins were subjected to SDS-PAGE electrophoresis. Each gel slice underwent in-gel digestion for protein extraction. The slices were diced into 1 mm pieces and then treated with 10 mM tris(2-carboxyethyl)phosphine hydrochloride (SIGMA) at 56 °C for 30 min for reduction, followed by alkylation with 55 mM iodoacetamide at room temperature for 45 min in the dark. Subsequently, digestion was carried out using Trypsin (MS Grade, Thermo Scientific) at 37 °C for 16 h. The resulting peptides were extracted using 1% trifluoroacetic acid and 50% acetonitrile. Phosphorylated peptides were enriched either using the High-Select Fe-NTA Phosphopeptide Enrichment Kit (Thermo Scientific) or the Titansphere Phos-TiO Kit (GL Science) following the manufacturer's instructions. Both the trypsin digest before enrichment and the enriched fractions were desalted using in-house C18 stage-tip and then used for LC-MS/MS analysis.

Mass spectra were acquired using a Thermo Scientific LTQ-Orbitrap Velos Pro connected to a nano-flow UHPLC system (ADVANCE UHPLC; AMR Inc.) with an Advanced Captive Spray SOURCE (AMR Inc.). Peptide mixtures were injected onto a C18 trap column (PepMap Neo Trap Cartridge, ID 0.3 mm × 5 mm, particle size 5 μm, Thermo Fisher Scientific) and subsequently fractionated by C18 reverse-phase chromatography (3 μm, ID 0.075 mm × 150 mm, CERI). Peptides were eluted with a linear gradient of solvent B (5–35% acetonitrile, 0.1% formic acid) at a flow rate of 300 nL/min over 60 min. The mass spectrometer performed seven successive scans, starting with a full MS scan from 350 to 1600 $m/z$ using Orbitrap (resolution = 60,000), followed by data-dependent scans of the top three most abundant ions using CID in the ion trap for the second to fourth scans, and using HCD in the Orbitrap (resolution = 7500) for the fifth to seventh scans. Automatic MS/MS spectra were acquired from the highest peak in each scan, with a relative collision energy set to 35% CID or HCD and an exclusion time of 90 s for ions within the same $m/z$ range.

The raw files were searched against the Uniprot *Mus musculus* proteome database (downloaded January 2022) and cRAP contaminant proteins dataset using the MASCOT program (version 2.6; Matrix Science) via Proteome Discoverer 2.5 (Thermo Fisher Scientific). The search was conducted with carbamidomethylation of cysteine as a fixed modification, and oxidation of methionine, acetylation of protein N-termini, and phosphorylation of serine, threonine, and tyrosine as variable modifications. The number of missed cleavage sites was set to 2.

### Statistical analysis

Graphing and statistical analysis were performed using Excel, R, and GraphPad Prism. Statistical tests used were described in figure legends. Student's *t* test was used when the data showed normal distribution. No sample size estimate was performed. No blinding was performed.

## Data availability

This study includes no data deposited in external repositories.

The source data of this paper are collected in the following database record: biostudies:S-SCDT-10_1038-S44318-025-00461-w.

## Peer review information

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

## Acknowledgements

We thank K Tanaka and M Ikeda for instructing protein expression and purification; K Aoki for providing Eevee-spCDK; R. Benezra and H Shibuya for providing NDC80 antibodies; J Ellenberg for providing macros for automated

microscopy; and the imaging, genome analysis, and animal facilities of RIKEN Kobe for technical support. We also thank our laboratory members for the discussion. This work was supported by RIKEN intramural grants, RIKEN Pioneering Project "Long-timescale Molecular Chronobiology", JSPS KAKENHI JP18H05549/JP21H02407/23H04948/25H00981 to TSK, and JSPS KAKENHI JP17K15069/JP19K06682/JP22K06257 to SY.

## Author contributions

**Shuhei Yoshida**: Conceptualization; Formal analysis; Funding acquisition; Investigation; Writing—original draft; Writing—review and editing. **Reiko Nakagawa**: Investigation; Methodology; Writing—original draft. **Kohei Asai**: Formal analysis; Investigation; Writing—original draft. **Tomoya S Kitajima**: Conceptualization; Supervision; Funding acquisition; Writing—original draft; Project administration; Writing—review and editing.

Source data underlying figure panels in this paper may have individual authorship assigned. Where available, figure panel/source data authorship is listed in the following database record: biostudies:S-SCDT-10_1038-S44318-025-00461-w.

## Disclosure and competing interests statement

The authors declare no competing interests.

# Expanded View Figures

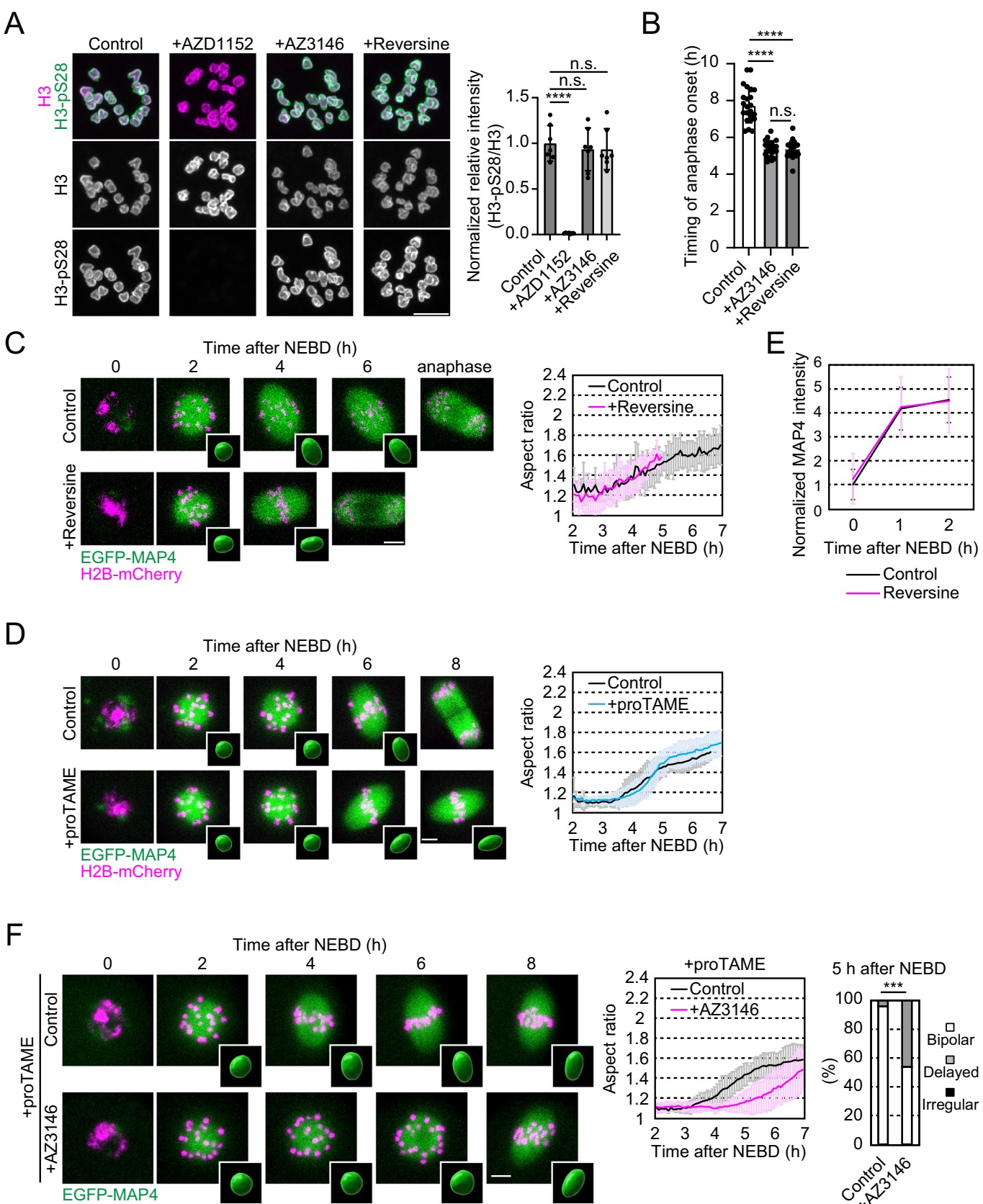

◄  **Figure EV1.  MPS1 activity is required for efficient spindle bipolarization.**

(A) Reversine or AZ3146 does not significantly inhibit Aurora B/C. Immunostaining of histone H3, H3-pS28 and Hoechst33342 (DNA) in oocytes treated with AZD1152 (Aurora B/C inhibitor), AZ3146, or reversine. Drugs were added to oocyte culture immediately after induction of meiotic resumption. Oocytes were fixed at 2 h after NEBD. Normalized relative intensities of H3-pS28 are shown (mean ± SD, $n = 7, 7, 7, 7$ oocytes. Three independent experiments were performed). n.s. not significant, ****$P = 0.0000000005$ by Tukey's multiple comparison test. Note that H3-pS28, an Aurora B/C target, was significantly reduced by AZD1142, but not by AZ3146 or reversine. (B) Reversine and AZ3146 accelerate anaphase onset. Timing of anaphase onset in oocytes treated with AZ3146 or reversine are shown (mean ± SD, $n = 23, 23, 23$ oocytes from three independent experiments). n.s. not significant, ****$P < 0.0000000001$ by Tukey's multiple comparison test. Oocytes treated with reversine or AZ3146 exhibited accelerated anaphase onset, consistent with spindle checkpoint defects caused by MPS1 inhibition (Hached et al, 2011; Yakoubi et al, 2017). (C) MPS1 inhibition does not delay spindle bipolarization but prematurely show anaphase spindle elongation. Live imaging of oocytes treated with reversine. Temporal changes in the aspect ratio of the spindle are shown (mean ± SD, $n = 8, 8$ oocytes). (D) ProTAME does not impair spindle bipolarization. Live imaging of oocytes treated with proTAME. Temporal changes in the aspect ratio of the spindle are shown (mean ± SD, $n = 9, 10$ oocytes. Three independent experiments were performed). (E) Initial microtubule nucleation is not significantly affected by MPS1 inhibition. Live imaging was performed on oocytes treated with proTAME and reversine (images are shown in Fig. 1B). Normalized EGFP-MAP4 intensities are shown (mean ± SD, $n = 26, 26$ oocytes from 4 independent experiments). (F) MPS1 inhibition delays spindle bipolarization in proTAME-treated oocytes. Live imaging of oocytes with proTAME and the MPS1 inhibitor AZ3146. Temporal changes in the aspect ratio of the spindle (mean ± SD, $n = 8, 8$ oocytes) and morphology classification at 5 h after NEBD ($n = 25, 26$ from 4 independent experiments) are shown. ***$P = 0.0008$ by Fisher's exact test for "bipolar" groups. Scale bars, 10 µm. Source data are available online for this figure.

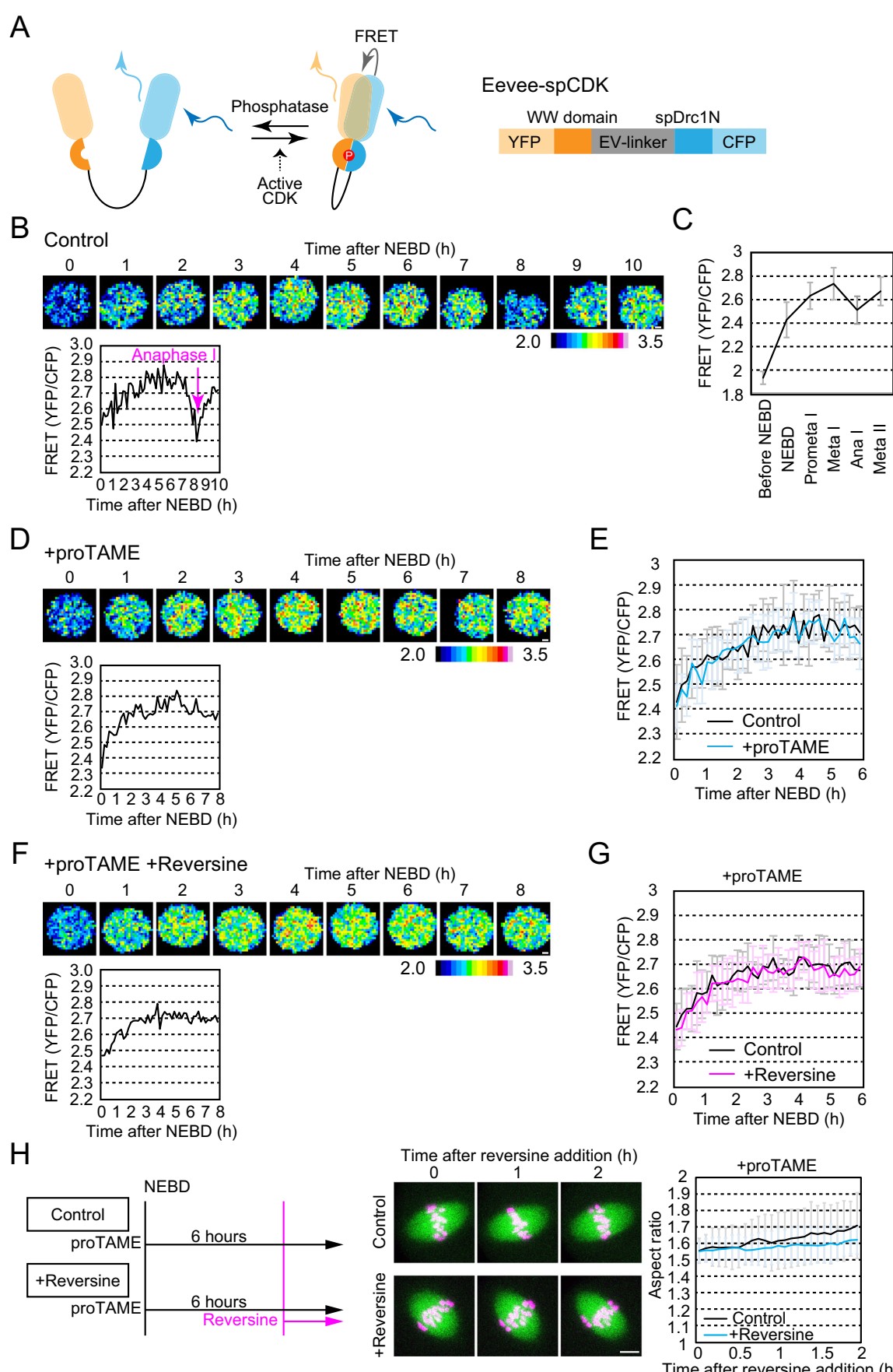

◄

**Figure EV2.   ProTAME or additional reversine treatment does not affect CDK1 activity dynamics during prometaphase and metaphase.**

(A) Schematic illustration of the CDK biosensor Eevee-SpCDK. (B, C) Temporal dynamics of the CDK1 activity. The CDK1 activity sensor Eevee-spCDK was monitored with live imaging. FRET (YFP/CFP) ratio images are shown. The plot in (B) shows the cytoplasmic FRET ratio in an oocyte. The plot in C shows the cytoplasmic FRET ratio at "Before NEBD" (0.5 h before NEBD), "NEBD", "Prometa I" (2 h after NEBD), "Meta I" (1 h before anaphase onset), "Ana I" (1 h after anaphase onset), and "Meta II" (3 h after anaphase onset) (mean ± SD, $n = 9$ oocytes from three independent experiments). Note that the Eevee-spCDK FRET dynamics is consistent with the CDK1 activity dynamics that increases upon M-phase entry (NEBD), gradually increases through prometaphase I and metaphase I, decreases at anaphase I, and then increases again at metaphase II (Choi et al, 1991; Davydenko et al, 2013). (D, E) CDK1 activity dynamics in proTAME-treated oocytes. FRET (YFP/CFP) ratio images are shown. The plots show the data from an oocyte (in D) and multiple oocytes (in E, mean ± SD, $n = 9$, 9 oocytes). Three independent experiments were performed. (F, G) CDK1 activity dynamics in oocytes treated with proTAME and reversine. Images and plots are shown as in (D, E) (mean ± SD, $n = 12$, 12 oocytes). Three independent experiments were performed. (H) MPS1 is not required for bipolar spindle maintenance. Live imaging of oocytes expressing EGFP-MAP4 (microtubules, green) and H2B-mCherry (chromosomes, magenta). Oocytes were cultured in the presence of proTAME. Reversine was added at 6 h after NEBD (metaphase I). Temporal changes in the aspect ratio of the spindle after reversine addition (mean ± SD, $n = 15$, 13 oocytes from 3 independent experiments) are shown. Scale bars, 10 µm.

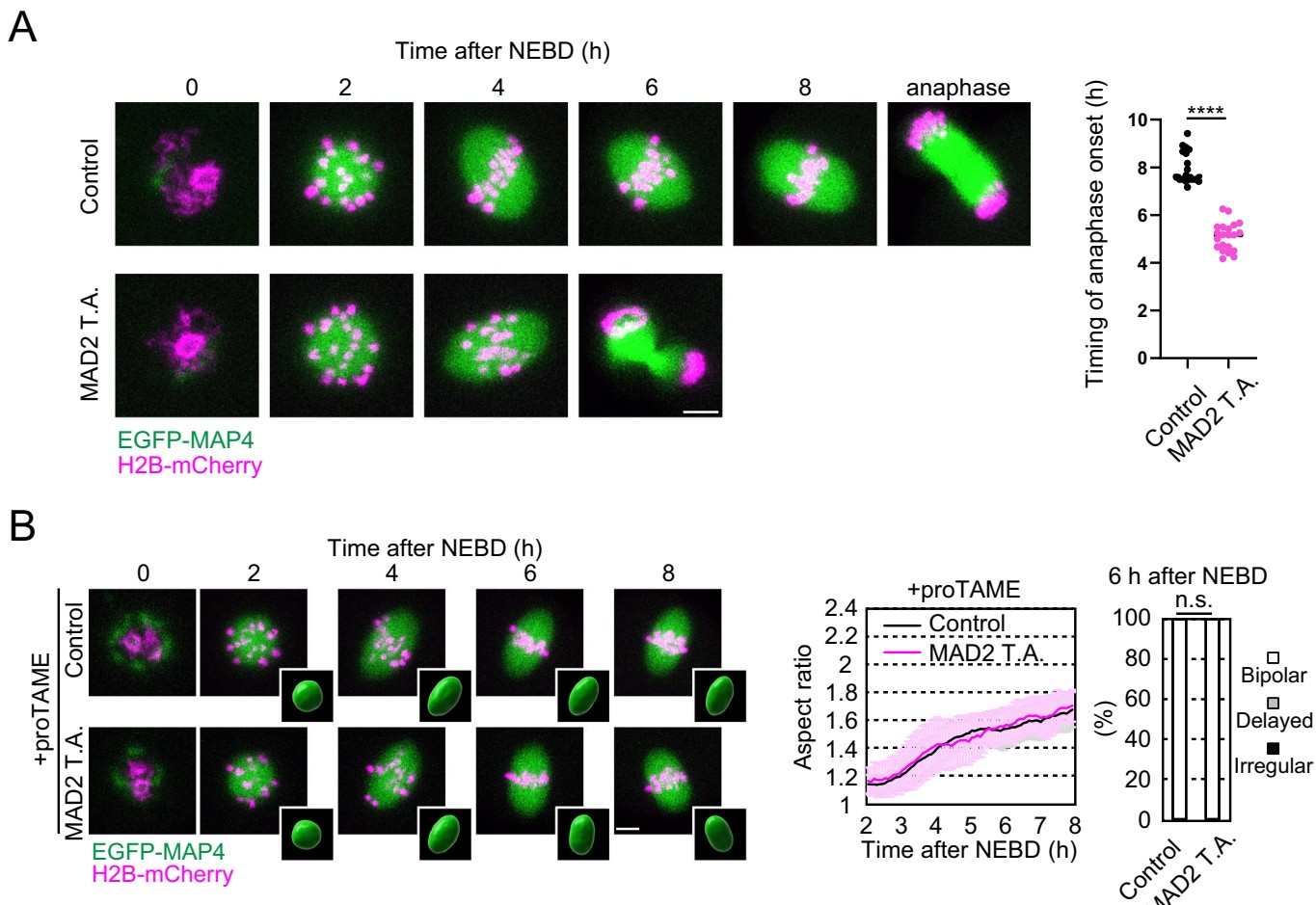

**Figure EV3. Defective spindle checkpoint does not delay spindle bipolarization.**

(A) MAD2 TRIM-Away accelerates anaphase onset in mouse oocytes. Representative z-projection images of EGFP-MAP4 (spindle, green) and H2B-mCherry (chromosome, magenta) are shown. MAD2 antibody was co-injected with TRIM21 mRNA for MAD2 TRIM-Away (T.A.). Timing of anaphase onset after NEBD are shown ($n = 18$, 21 oocytes from 3 independent experiments). ****$P = 0.00000000003$ by two-tailed unpaired Mann–Whitney test. (B) MAD2 is not required for spindle bipolarization in mouse oocytes. Live imaging of oocytes with proTAME. Temporal changes in the aspect ratio of the spindle (mean ± SD, $n = 14$, 14 oocytes) and morphology classification at 6 h after NEBD ($n = 27$, 28 oocytes) are shown. Four independent experiments were performed. n.s., not significant by Fisher's exact test for "bipolar" groups. Scale bars, 10 μm.

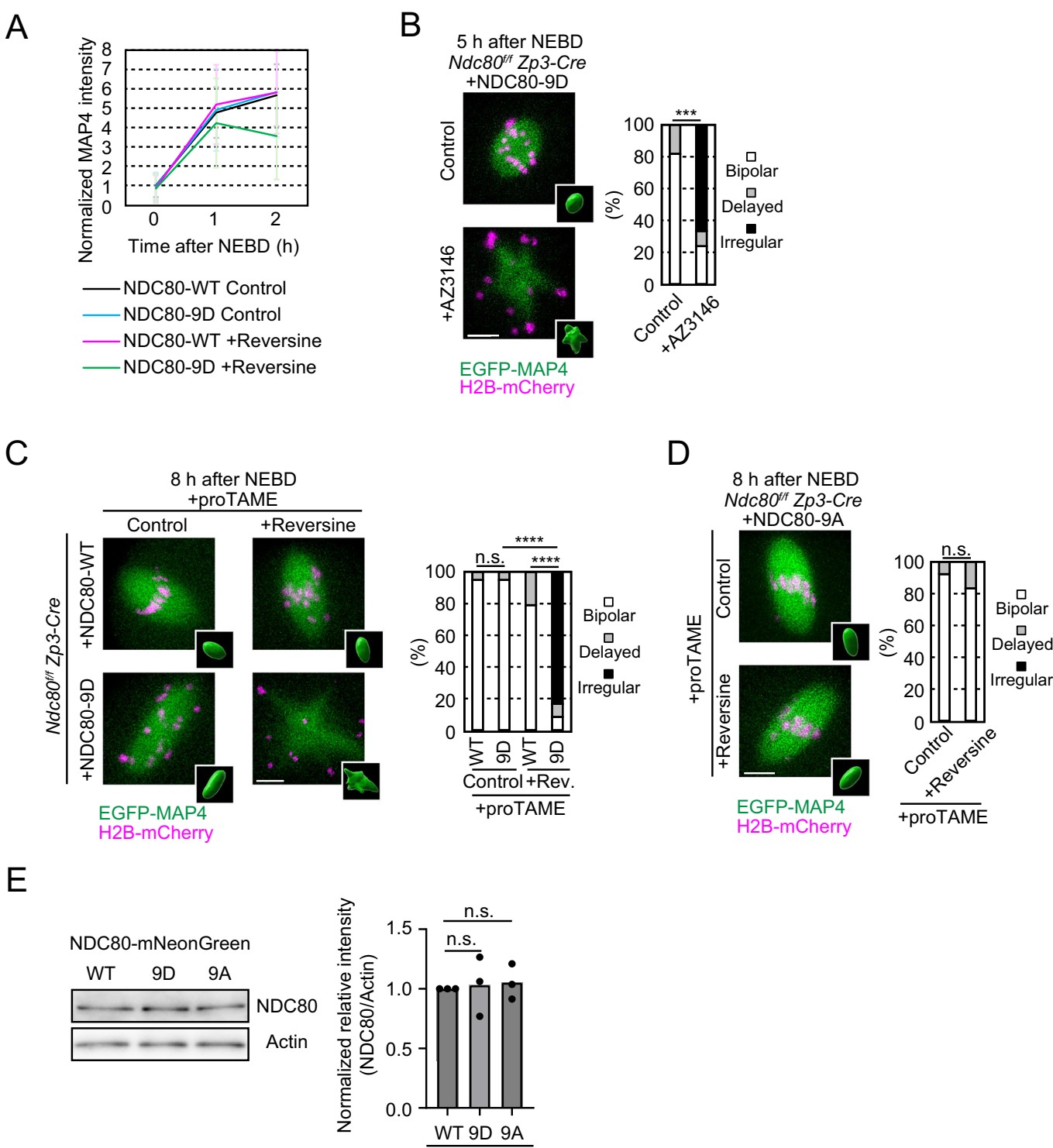

◀ **Figure EV4. MPS1 activity is required for spindle bipolarization in the absence of stable kinetochore–microtubule attachment.**

(A) MPS1 is required for microtubule maintenance in the absence of stable kinetochore–microtubule attachments. Live imaging was performed on *Ndc80^{f/f} Zp3-Cre* oocytes expressing NDC80-WT/-9D and treated with reversine, using the microtubule marker EGFP-MAP4 (images are shown in Fig. 1C, D). Normalized EGFP-MAP4 intensities are shown (mean ± SD, $n = 21, 25, 25, 23$ from 3 independent experiments) are shown. (B) MPS1 inhibition impairs spindle bipolarization in NDC80-9D oocytes. Live imaging of *Ndc80^{f/f} Zp3-Cre* oocytes expressing EGFP-MAP4 (spindle, green), H2B-mCherry (chromosome, magenta) and NDC80-9D with the MPS1 inhibitor AZ3146. Z-projection and 3D-reconstruction images at 5 h after NEBD are shown. Spindle morphology classification is shown ($n = 22, 21$ oocytes from three independent experiments). ***$P = 0.0002$. by Fisher's exact test for "bipolar" groups. (C) MPS1 inhibition impairs spindle bipolarization in NDC80-9D oocytes even after metaphase arrest. Live imaging of *Ndc80^{f/f} Zp3-Cre* oocytes expressing NDC80-WT/-9D in the presence of proTAME. Z-projection and 3D-reconstruction images at 8 h after NEBD are shown. *Ndc80^{f/f} Zp3-Cre* oocytes expressing NDC80-WT/-9D were analyzed as in (B) ($n = 20, 20, 24, 24$ oocytes from three independent experiments). n.s. not significant, ****$P = 0.000000010$ (9D control vs 9D revesine), $0.000001$ (WT reversine vs 9D reversine) by Fisher's exact test for "bipolar" groups. (D) MPS1 inhibition does not impair spindle bipolarization in NDC80-9A oocytes. Z-projection and 3D-reconstruction images at 8 h after NEBD are shown. *Ndc80^{f/f} Zp3-Cre* oocytes expressing NDC80-9A were analyzed as in (B) ($n = 26, 24$ oocytes from 3 independent experiments). n.s., not significant by Fisher's exact test for "bipolar" groups. (E) Protein levels of the mutant forms of NDC80. Oocytes at the GV stage were used for the Western blotting of NDC80-WT/9D/9A-mNeonGreen and actin. Normalized relative intensities from three independent experiments are shown. n.s., not significant by Tukey's multiple comparison test.

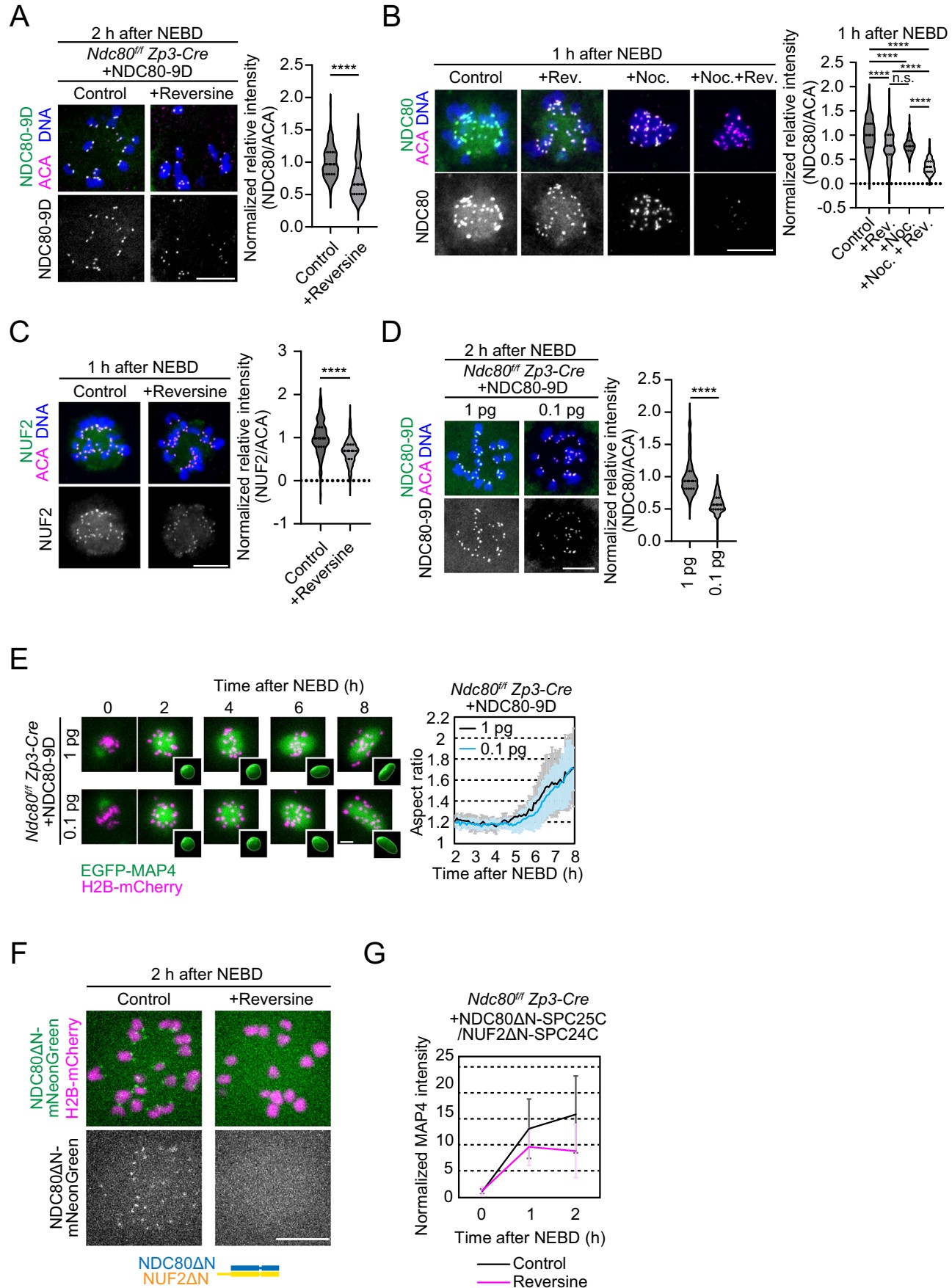

◀ **Figure EV5. MPS1 promotes kinetochore localization of NDC80 and NUF2.**

(A) MPS1 inhibition decreases NDC80-9D localization at kinetochores. Immunostaining of NDC80-9D-GFP, ACA (kinetochores), and Hoechst33342 (DNA) in oocytes treated with reversine. Normalized relative intensities of NDC80-9D-GFP are shown (median and quartiles, $n = 280$, 174 kinetochores from 7, 7 oocytes). Three independent experiments were performed. ****$P < 0.0000000001$ by two-tailed unpaired Mann–Whitney test. (B) MPS1 inhibition decreases NDC80 localization at kinetochores. Immunostaining of anti-NDC80, ACA (kinetochores), and Hoechst33342 (DNA) in oocytes treated with reversine and/or nocodazole at 1 h after NEBD. Normalized relative intensities of NDC80 are shown (median and quartiles, $n = 240$, 240, 240, 232 kinetochores from 6, 6, 6, 6 oocytes). Three independent experiments were performed. n.s. not significant, ****$P < 0.0000000001$ by Tukey's multiple comparison test. (C) MPS1 inhibition decreases NUF2 localization at kinetochores. Immunostaining of NUF2-HA, ACA (kinetochores), and Hoechst33342 (DNA) in oocytes treated with reversine. Normalized relative intensities of NUF2-HA are shown (median and quartiles, $n = 280$, 280 kinetochores from 7, 7 oocytes). Three independent experiments were performed. ****$P < 0.0000000001$ by two-tailed unpaired Student's $t$ test. (D) Titration of NDC80-9D. Immunostaining of NDC80-9D-GFP, ACA (kinetochores) and Hoechst33342 (DNA) in oocytes expressing 1 pg or 0.1 pg mRNA of NDC80-9D-GFP. Normalized relative intensities of NDC80-9D-GFP are shown (median and quartiles, $n = 280$, 200 kinetochores of 7, 5 oocytes). Two independent experiments were performed. ****$P < 0.0000000001$ by two-tailed unpaired Mann–Whitney test. (E) Reduced NDC80-9D can support spindle bipolarization. Live imaging of $Ndc80^{f/f}$ Zp3-Cre oocytes expressing EGFP-MAP4 (spindle, green), H2B-mCherry (chromosome, magenta), and NDC80-9D. Temporal changes in the aspect ratio of the spindle are shown (mean ± SD, $n = 6$, 7 oocytes). Three independent experiments were performed. (F) MPS1 inhibition delocalizes NDC80ΔN from kinetochores. $Ndc80^{f/f}$ Zp3-Cre oocytes expressing NDC80ΔN-mNeonGreen, NUF2ΔN, and H2B-mCherry treated with reversine were imaged. Three independent experiments were performed. (G) MPS1 is required for microtubule maintenance in the absence of stable kinetochore–microtubule attachments. Live imaging was performed on $Ndc80^{f/f}$ Zp3-Cre oocytes expressing NDC80ΔN-SPC25C and NUF2ΔN-SPC24C and treated with reversine, using the microtubule marker EGFP-MAP4 (images are shown in Fig. 2D). Normalized EGFP-MAP4 intensities are shown (mean ± SD, $n = 26$, 26 from three independent experiments). The experiment was performed in the presence of proTAME. Scale bars, 10 μm.

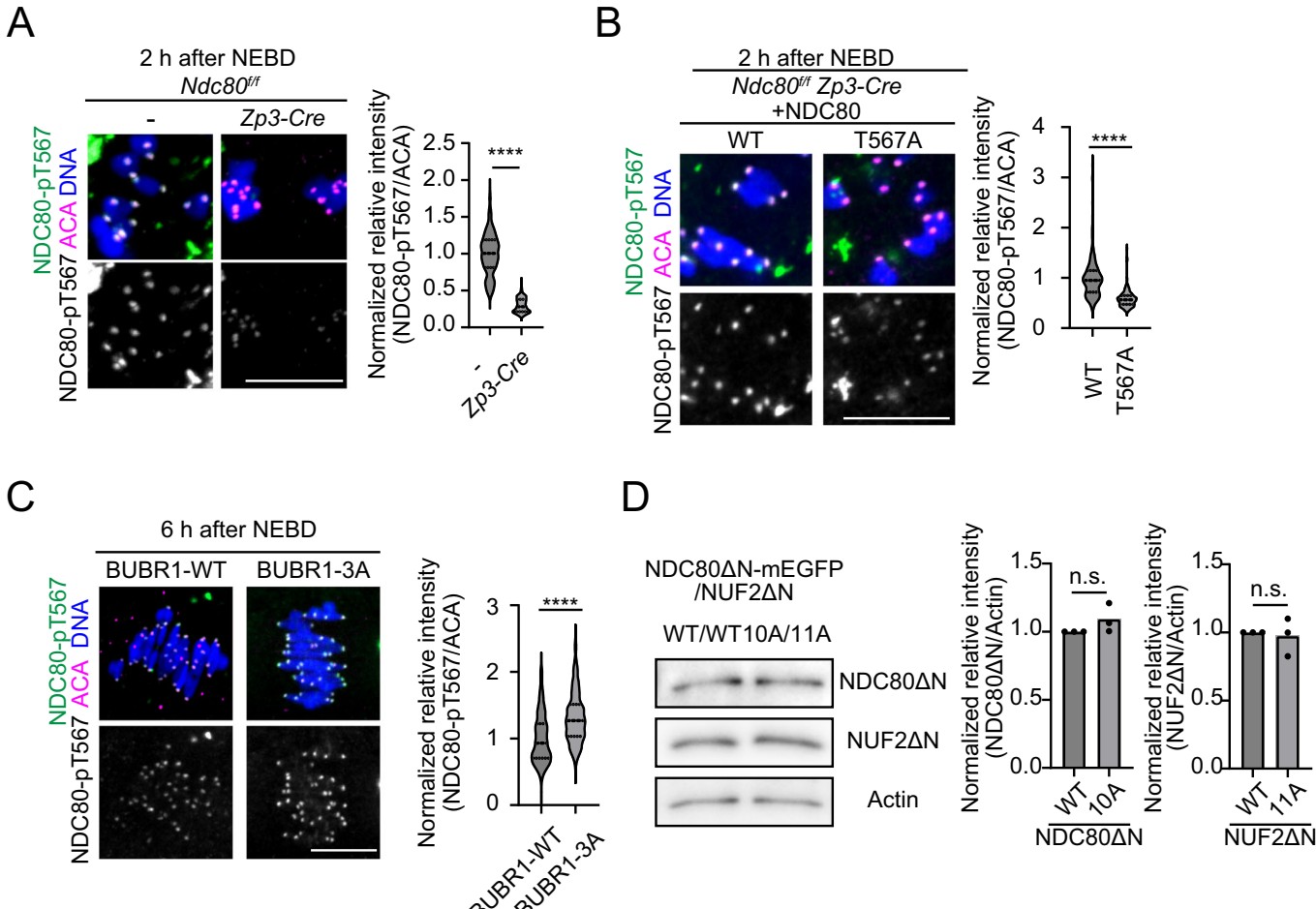

**Figure EV6. MPS1 promotes spindle bipolarization via the C-terminal domains of NDC80-NUF2 during prometaphase.**

(A) Phospho-antibody specificity. *Ndc80^{f/f} Zp3-Cre* oocytes were immunostained with anti-phospho-NDC80-T567 antibody, ACA (kinetochores), and Hoechst33342 (DNA). Control oocytes were without *Zp3-Cre*. Normalized relative intensities of phospho-NDC80-T567 are shown (median and quartiles, $n = 199$, 168 kinetochores from 5, 5 oocytes). Three independent experiments were performed. ****$P < 0.0000000001$ by two-tailed unpaired Mann–Whitney test. (B) Phospho-antibody specificity. *Ndc80^{f/f} Zp3-Cre* oocytes expressing NDC80-WT/-T567A were immunostained with anti-phospho-NDC80-T567, ACA (kinetochores), and Hoechst33342 (DNA). Normalized relative intensities of NDC80-pT567 are shown (median and quartiles, $n = 240$, 240 kinetochores from 6, 6 oocytes). Three independent experiments were performed. ****$P < 0.0000000001$ by two-tailed unpaired Mann–Whitney test. (C) NDC80-T567 phosphorylation increases in BUBR1-3A-overexpressing oocytes. Oocytes expressing mEGFP-BUBR1-WT/3 A were immunostained with anti-phosphorylated NDC80 at T567 (NDC80-pT567), ACA (kinetochores), and Hoechst33342 (DNA). Normalized relative intensities of NDC80-pT567 are shown (median and quartiles, $n = 240$, 240 kinetochores from 6, 6 oocytes). Three independent experiments were performed. ****$<0.0000000001$ by two-tailed unpaired Mann–Whitney test. (D) Protein levels of the mutant forms of NDC80ΔN and NUF2ΔN. Oocytes at the GV stage were used for the Western blotting of NDC80ΔN-WT/10A-mEGFP, NUF2ΔN-WT/11 A and actin. Normalized relative intensities from three independent experiments are shown. n.s., not significant by two-tailed unpaired Mann–Whitney test. Scale bars, 10 μm.

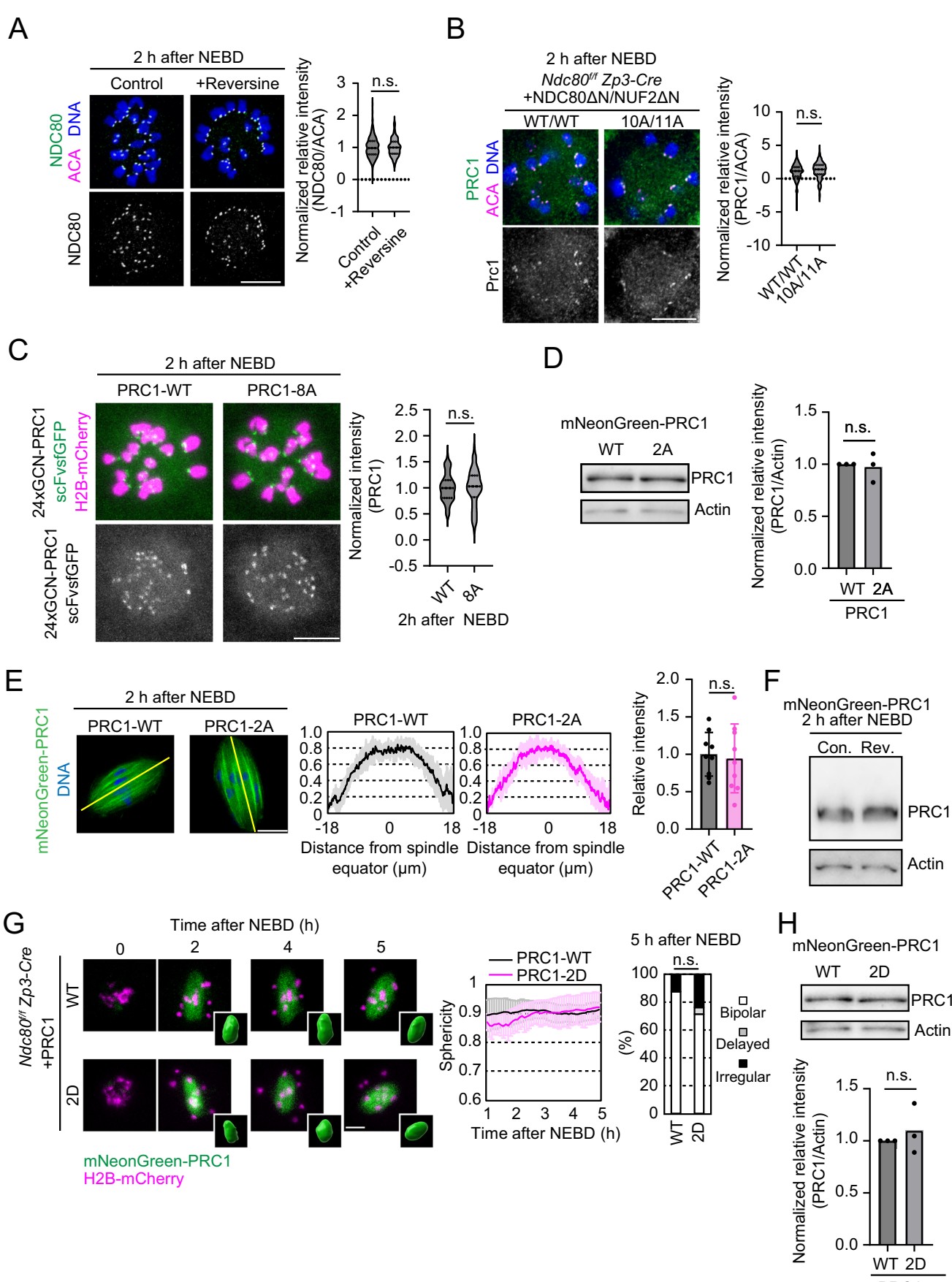

◀  **Figure EV7.   MPS1 activity promotes kinetochore localization and spindle bipolarization activity of PRC1.**

(A) NDC80 is not significantly decreased in MPS1-inhibited oocytes at 2 h after NEBD. Immunostaining of reversine-treated oocytes with anti-NDC80, ACA (kinetochores), and Hoechst33342 (DNA). Normalized relative intensities of NDC80 are shown (median and quartiles, $n = 240$, 240 kinetochores from 6, 6 oocytes). Three independent experiments were performed. n.s., not significant by two-tailed unpaired Student's t test. (B) PRC1 can be recruited to kinetochores by phospho-mutant NDC80ΔN and NUF2ΔN. Immunostaining of *Ndc80^{f/f} Zp3-Cre* oocytes expressing NDC80ΔN-WT/-10A and NUF2ΔN-WT/-11A with anti-PRC1, ACA (kinetochores), and Hoechst33342 (DNA). Normalized relative intensities of PRC1 are shown (median and quartiles, $n = 200$, 200 kinetochores from 5, 5 oocytes). Three independent experiments were performed. n.s., by two-tailed unpaired Student's t test. (C) Phospho-mutant PRC1 can localize to kinetochores. Oocytes expressing 24xGCN-PRC1-WT/-8A, scFv-sfGFP, and H2B-mCherry were imaged. Relative intensities of PRC1-WT or -8A at kinetochores are shown (median and quartiles, $n = 80$, 80 kinetochores from 4, 4 oocytes). Three independent experiments were performed. n.s., not significant by two-tailed unpaired Mann–Whitney test. (D) Protein levels of the mutant forms of PRC1. Oocytes at the GV stage were used for the Western blotting of mNeonGreen-PRC1-WT/2 A and actin. Normalized relative intensities from 3 independent experiments are shown. n.s., not significant by two-tailed unpaired Mann–Whitney test. (E) PRC1-2A preferentially localizes to the middle region of the spindle. Oocytes expressing mNeonGreen-PRC1-WT/2 A were fixed. Images of mNeonGreen-PRC1 and Hoechst33342 (DNA) are shown. In each oocyte, intensities of PRC1-WT or −2A along the spindle axis (yellow line) were normalized with their maximum intensity (mean ± SD, $n = 9$, 9 oocytes from three independent experiments). PRC1 intensities on the spindle were compared between PRC1-WT and −2A. n.s., not significant by two-tailed unpaired Mann–Whitney test. (F) No detectable MPS1-dependent band shifts of PRC1 in oocyte extracts. Oocytes expressing mNeonGreen-PRC1 were cultured in the presence of reversine. Oocytes at prometaphase (2 h after NEBD) were used for Phos-tag SDS-PAGE followed by Western blotting. (G) PRC1-2D can rescue spindle bipolarization in *Ndc80*-deleted oocytes. Live imaging of *Ndc80^{f/f} Zp3-Cre* oocytes expressing mNeonGreen-PRC1-WT/2D (T578 and S583 substituted to aspartic acid, green) and H2B-mCherry (chromosome, magenta). Insets show 3D reconstructed images. Temporal changes in the sphericity of the spindle (mean ± SD, $n = 14$, 13 oocytes from two independent experiments) and morphology classification at 5 h after NEBD ($n = 24$, 21 oocytes from three independent experiments) are shown. n.s., not significant by Fisher's exact test for "bipolar" groups. (H) Protein levels of the mutant forms of PRC1. Oocytes at the GV stage were used for the Western blotting of mNeonGreen-PRC1-WT/2D and actin. Normalized relative intensities from three independent experiments are shown. n.s., not significant by two-tailed unpaired Mann–Whitney test. Scale bars, 10 μm.

A

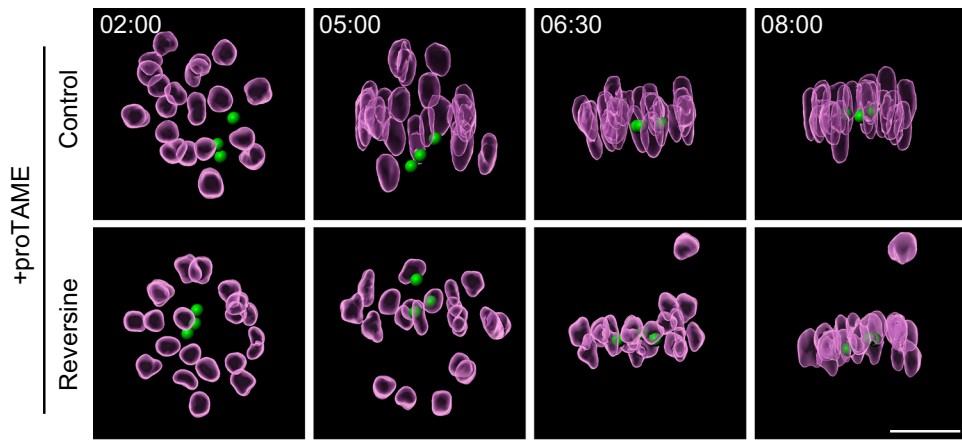

NDC80-NUF2 beads / H2B-mCherry

B

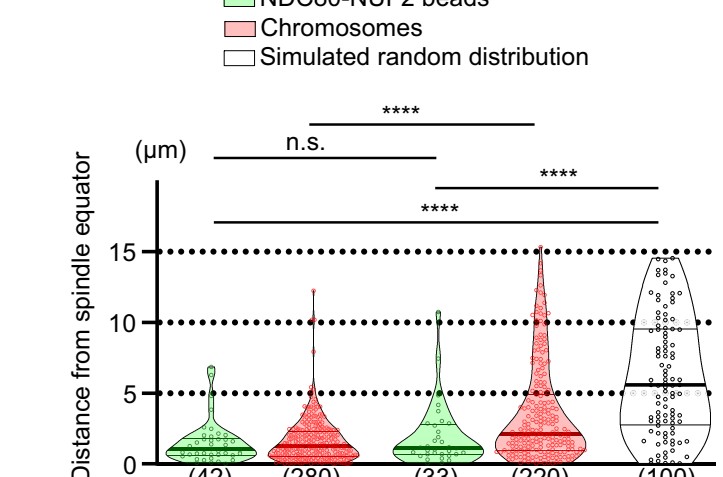

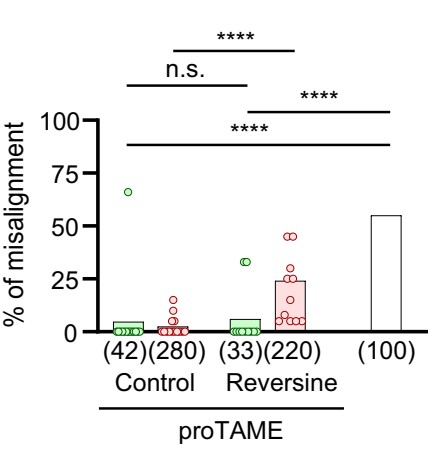

C

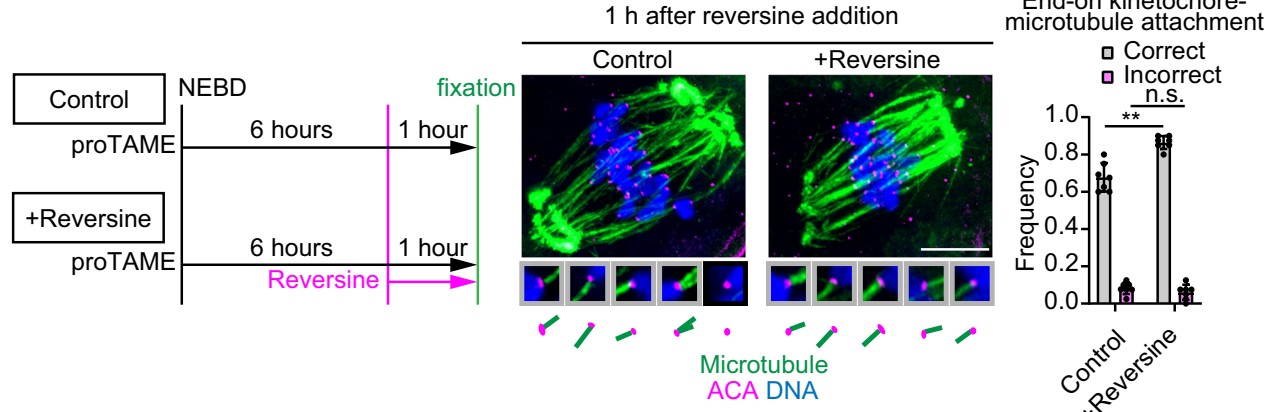

Microtubule
ACA DNA

**Figure EV8. MPS1 is not required for NDC80-NUF2 microbead alignment.**

(A) Live imaging of NDC80-NUF2-tethered beads in oocytes. Oocytes expressing NDC80-GFP, NUF2 and H2B-mCherry and carrying anti-GFP beads were cultured for meiosis I in the presence of proTAME and reversine. 3D reconstructed images of chromosomes (H2B-mCherry, magenta) and NDC80-NUF2 beads (NDC80-GFP, green) are shown. Time after NEBD (hours:minutes). (B) NDC80-NUF2 beads can align in MPS1-inhibited oocytes. (Left) The distances of NDC80-NUF2 beads and chromosomes from the spindle equator are shown ($n = 42$, 33 beads, 280, 220 chromosomes from 14, 11 oocytes in one of three experiments). Simulated random distribution of particles within a spindle-like ellipsoid (30 μm in length and 20 μm in width) is used as a reference. n.s., not significant, ****$P = 0.00000000169$ (control chromosome vs reversine chromosome), 0.00000000002 (control NDC80-NUF2 beads vs simulated random distribution), 0.00000013011 (reversine NDC80-NUF2 beads vs simulated random distribution) by Kruskal-Wallis test with Dunn's correction. (Right) Fraction of misaligned NDC80-NUF2 beads and chromosomes are shown. Beads and chromosomes positions >5 μm from the spindle equator are defined as "misalignment". n.s., not significant. ****$P = 0.000000000000039$ (control chromosome vs reversine chromosome), 0.0000000040 (control NDC80-NUF2 beads vs simulated random distribution), 0.00000019 (reversine NDC80-NUF2 beads vs simulated random distribution) by Fisher's exact test. (C) MPS1 inhibition after metaphase spindle establishment does not increase kinetochore–microtubule attachment errors. Oocytes were cultured in the presence of proTAME. Reversine was added at 6 h after NEBD (metaphase I). Oocytes were collected 1 h after the reversine addition, treated briefly with a cold buffer, and then immunostained for stable microtubules (green), kinetochores (magenta), and DNA (blue). Magnified images of end-on monopolar attachments (correct, gray frame) are shown (mean ± SD, $n = 7$, 7 oocytes from three independent experiments). n.s., not significant, **$P = 0.001166$ by two-tailed unpaired Mann–Whitney test. Scale bars, 10 μm.

