## [Peer Review File · The EMBO Journal]

MPS1 promotes timely spindle bipolarization to prevent kinetochore-microtubule attachment errors in oocytes

Shuhei Yoshida, Reiko Nakagawa, Kohei Asai, and Tomoya Kitajima

Corresponding author(s): Tomoya Kitajima (tomoya.kitajima@riken.jp), Shuhei Yoshida (shuhei.yoshida@riken.jp)

Review Timeline:

Submission Date:	2nd Sep 24
Editorial Decision:	1st Oct 24
Revision Received:	26th Mar 25
Editorial Decision:	22nd Apr 25
Revision Received:	23rd Apr 25
Accepted:	25th Apr 25

Editor: Hartmut Vodermaier

Transaction Report:

Dr. Tomoya S Kitajima
RIKEN Center for Biosystems Dynamics Research
Laboratory for Chromosome Segregation
2-2-3 Minatojima-minamimachi, Chuo-ku
Kobe
650-0047
Japan

1st Oct 2024

Re: EMBOJ-2024-118908
MPS1 promotes timely spindle bipolarization to prevent kinetochore-microtubule attachment errors in oocytes

Dear Dr. Kitajima,

Thank you for submitting your study on MPS1 roles in facilitating oocyte spindle bipolarization to The EMBO Journal. Three expert referees have now evaluated it and returned the comments copied below. As you will see, they all express interest in the work and appreciate its overall experimental. We would therefore be happy to consider a revised version further for publication, pending adequate addressing of various specific concerns and suggestions noted in the three reports.

It is our policy to consider only a single round of major revision and therefore important to fully respond to all comments at the time of resubmission - so please do not hesitate to get back to me with a tentative response letter/revision plan, in case you would like to clarify/discuss certain points and how they might be answered already during the early stages of the revision. I should add that we could also offer extension of the default three-months revision period if needed, with our 'scooping protection' (meaning that competing work appearing elsewhere in the meantime will not affect our considerations of your study) remaining of course valid also throughout this extension.

Detailed information on preparing, formatting and uploading a revised manuscript can be found below and in our Guide to Authors. Thank you again for the opportunity to consider this work for The EMBO Journal, and I look forward to your revision in due time.

Yours sincerely,

Hartmut Vodermaier

9) To facilitate reproducibility and cross-laboratory adoption of methodologies, please structure the Materials & Methods section as outlined in our guide to authors, including a completed Reagents and Tools Table that can be downloaded from our author guidelines as well (<https://www.embopress.org/page/journal/14602075/authorguide#structuredmethods>).

10) Digital image enhancement is acceptable practice, as long as it accurately represents the original data and conforms to community standards. If a figure has been subjected to significant electronic manipulation, this must be clearly noted in the figure legend and/or the 'Materials and Methods' section. The editors reserve the right to request original versions of figures and the original images that were used to assemble the figure. Finally, we generally encourage uploading of numerical as well as gel/blot image source data; for details see: embopress.org/page/journal/14602075/authorguide#sourcedata

At EMBO Press, we ask authors to provide source data for the main manuscript figures. Our source data coordinator will contact you to discuss which figure panels we would need source data for and will also provide you with helpful tips on how to upload and organize the files.

In the interest of ensuring the conceptual advance provided by the work, we recommend submitting a revision within 3 months (30th Dec 2024). Please discuss the revision progress ahead of this time with the editor if you require more time to complete the revisions. Use the link below to submit your revision:

Link Not Available

Referee #1:

In the absence of canonical centrosomes, meiotic spindle bipolarization in mouse and human oocytes is a gradual process that unfolds over several hours. This study investigates the role of kinetochore-associated MPS1 activity in facilitating spindle bipolarization. Through a series of carefully designed experiments, the authors propose a model where MPS1 at kinetochores phosphorylates key substrates, including Ndc80, Nuf2, and PRC1, which are crucial for timely spindle bipolarization, particularly when microtubule-kinetochore attachments are unstable. The study is well-executed and sheds light on an important aspect of oocyte chromosome segregation.

I only have two minor comments which might further strengthen the findings presented.

Minor comments:

1/ the authors should compare the expression levels of all the Ndc80, Nuf2, and PRC1 mutants introduced via cRNA injections to ensure they are comparable.

2/ could the authors compare the amount of microtubules at BD +2h in all the conditions where bipolarization is delayed? My impression is that there are less microtubules around chromosomes in all these conditions (Fig 1D, 2D, 3C & D). If this is the case, could they check whether it is the nucleation and/or stability of microtubules that is affected in these conditions. This could be an interesting observation to further understand and discuss the mechanism by which timely spindle bipolarization occurs in acentrosomal meiotic spindles. Such an observation could also potentially slightly modify their interpretation as well as their

model presented in Fig. 5.

Referee #2:

The accurate segregation of chromosomes depends on spindle biorientation, where each kinetochore pair forms a bipolar attachment to the spindle. In oocytes, which lack centrosomes, kinetochores are initially encircled by randomly oriented microtubules, often resulting in incorrect attachments and potential chromosome segregation errors. Despite this, the factors responsible for initiating spindle biorientation have remained elusive. This study proposes that the kinase MPS1 plays a pivotal role in facilitating spindle bipolarization prior to stabilizing these erroneous attachments, through the regulation of key components such as NDC80, NUF2, and PRC1. In the absence of MPS1, spindle bipolarization is delayed, leading to defective microtubule attachments. The authors introduce a two-step model for spindle assembly in oocytes: In the initial phase, kinetochores with unstable attachments rely on MPS1 to recruit microtubule regulators, aiding in the formation of a bipolar spindle. In the second phase, stable attachments complete spindle assembly and prevent errors. In the first phase, MPS1 is suggested to regulate the proper kinetochore localization of NDC80 and NUF2, phosphorylate their C-terminal regions, and control PRC1 activity, establishing MPS1 as a critical factor in spindle bipolarization. The manuscript is well-constructed, with carefully designed and elegantly executed experiments. This is an exciting study and a good candidate for publication in EMBO. Below are some specific concerns:

Major Points:

In Figure 1b, the spindle is absent in reversine-treated oocytes compared to the control at "timepoint 0". This suggests that MPS1 inhibition by reversine delays spindle assembly. Could this delay explain the postponed bipolarization in reversine-treated oocytes?

Is this role in spindle bipolarization specific to MPS1, or could other spindle assembly checkpoint proteins also be involved?

The authors propose that MPS1 is crucial for spindle bipolarization. Previous studies have shown that both NDC80 and PRC1 are involved in this process. The authors suggest that MPS1 regulates spindle bipolarization through NDC80 and PRC1. Their results indicate that MPS1 affects the localization and phosphorylation of NDC80 at its C-terminal domain. Although phosphorylation is not required for bipolarization, the localization of the C-terminal domains of NDC80-NUF2 appears to act downstream of MPS1. However, the exact mechanism by which MPS1 regulates this process through NDC80-NUF2 remains unclear. Is it possible that MPS1 recruits PRC1 via the C-terminal domain of NDC80-NUF2?

Recently, the same group demonstrated that the NDC80-Nuf2 complex is sufficient to establish spindle biorientation (Asai K et al., 2024) using microbeads coated with NDC80-Nuf2 proteins. These microbeads were able to align at the metaphase plate independently of both the inner and outer kinetochore, as well as the inner centromere. I am wondering whether Mps1 is required to promote the biorientation of NDC80-Nuf2 microbeads?

Could the authors investigate how MPS1 regulates the localization of both NDC80 and PRC1 at the kinetochore?

In Figure S4B, the kinetochore staining of NDC80-T567 is weak but not completely absent in the NDC80 knockout. This raises questions about the specificity of the antibody used. The authors note that phosphorylation is reduced during metaphase I as MPS1 activity declines. How does this dephosphorylation occur, and which phosphatase might be responsible?

The authors show that PRC1 is a potential substrate of MPS1, based on in vitro experiments. Specifically, they identify two phosphorylation sites, T578 and S583, that may regulate spindle bipolarization downstream of MPS1. However, it remains to be confirmed if this occurs in vivo. Another intriguing question is how these phosphorylations regulate PRC1's function as an antiparallel microtubule cross-linker.

Minor Points:

The results in Figure S5C would be more convincing with quantification of PRC1-WT and PRC1-8A localization at the kinetochore.

Repeating the experiments at least three times and increasing the number of quantified oocytes will strengthen the conclusions.

Referee #3:

Yoshida et al. addresses an important question of spindle bipolarization relates to correct kinetochore-microtubule attachments in acentrosomal mice oocytes. They provide strong evidence that during prometaphase, when kinetochore-microtubule end-on attachments are not yet established, the MPS1 kinase initiates timely bipolarization of the acentrosomal spindle by promoting NDC80 and PRC1 recruitment to the kinetochore. The Cter of NDC80, but not the MPS1-mediated phosphorylation of the Cter domain, is sufficient for promoting spindle bipolarization. MPS1 also phosphorylates T578 and S583 on PRC1 to promote its spindle bipolarization activity. Interestingly, this study reveals that incorrect but stable kinetochore-microtubule attachments can promote spindle bipolarization, which in turn further stabilizes the kinetochore-microtubule attachments. Because error correction of incorrect kinetochore-microtubule attachments and the spindle assembly checkpoint surveillance is less stringent in acentrosomal oocytes, this is an important paper that elucidates how MPS1 acts through the NDC80-PRC1 pathway to initiate an early switch to spindle bipolarization to limit the number of attachment errors, thus lowering the initial number of incorrect attachments formed. The manuscript is well written and easy to follow. The findings reveal new mechanistic insight into the timing of spindle bipolarization and justifies how its temporal regulation improves chromosome alignment and protects against aneuploidy in meiosis.

Main points-

1. A reminder of the NDC80-9D and NDC80-9A mutants is required to properly understand the significance of Fig 1C and S3B-C. I cannot find the MDC80-9D mutant in the citations provided (Cheeseman et al., 2006; DeLuca et al., 2006) and any reference for NDC80-9A is missing.
2. Fig 2 claims that Mps1 promotes spindle bipolarization via Cter domains on both NDC80 and NUF2. While all experiments were conducted with both NDC80 and NUF2 mutants, the reduced localization is only assayed for NDC80. The current manuscript does not show any evidence of Mps1 regulating NUF2 localization at the kinetochore. The effects on NUF2 should also be shown in the experiments or the conclusions should be revised to not include NUF2.
3. A more detailed reminder of how KIFC/HSET accelerate spindle bipolarization would be helpful. Fig 4D reports improved chromosome alignment in MPS1-inhibited cells when KIF1C is overexpressed. It is important to clarify if this phenotype is because Kif1C directly regulates chromosome alignment in a MPS1-independent pathway or a result of spindle bipolarization.
4. In all relevant experiments, the timing when reversine is added to the cells should be included in the text/ figure legend, as this will clarify whether MPS1 activity promotes recruitment of NDC80/PRC1 to the kinetochore or does MPS1 activity prevent removal of NDC80/PRC1 from the kinetochore.
5. Is Mps1 is only required to initiate the spindle bipolarization or is it also required to maintain the bipolarized spindle? Adding reversine delays bipolarization. Does adding reversine after bipolarization reverse the bipolar spindle to a sphere-shaped spindle and/or weaken the stabilized kinetochore-microtubule attachments?
6. PRC1 overexpression rescues the spindle bipolarization in Ndc80-deleted oocytes without its kinetochore localization (Yoshida et al., 2020). However, MPS1 phosphorylation of PRC1 is required for PRC1's spindle bipolarization activity. Can the authors speculate on whether PRC1 needs to be at the kinetochore for it to get phosphorylated by MPS1 or do MPS1-PRC1 interact in other parts of the cell?
7. Is the MPS1-PRC1 pathway equally important for limiting kinetochore-microtubule attachment errors in metaphase II? A similar analysis as Fig 4B and 4E for meiosis II spindles will be very interesting. Looking at the role of MPS1 in the timing of spindle bipolarization in mitotic cells that have impaired error correction or spindle checkpoint will also reveal if MPS1 acts as a backup pathway for securing chromosome capture by microtubules before anaphase onset in centrosomal cells.

Point-by-point response to reviewer's comments

EMBOJ-2024-118908

MPS1 promotes timely spindle bipolarization to prevent kinetochore-microtubule attachment errors in oocytes

Correspondence to: tomoya.kitajima@riken.jp

Plain letters indicate reviewers' comments.

Blue letters indicate our responses.

"Red letters enclosed in quotation marks" indicate texts newly added to the revised manuscript.

Note: The line numbers shown for reference (**LXXX–XXX**) are from the Merged PDF file (not the Manuscript Text file).

First of all, we thank all reviewers for their suggestions and comments, which greatly improved our manuscript.

Referee #1:

In the absence of canonical centrosomes, meiotic spindle bipolarization in mouse and human oocytes is a gradual process that unfolds over several hours. This study investigates the role of kinetochore-associated MPS1 activity in facilitating spindle bipolarization. Through a series of carefully designed experiments, the authors propose a model where MPS1 at kinetochores phosphorylates key substrates, including Ndc80, Nuf2, and PRC1, which are crucial for timely spindle bipolarization, particularly when microtubule-kinetochore attachments are unstable. The study is well-executed and sheds light on an important aspect of oocyte chromosome segregation.

I only have two minor comments which might further strengthen the findings presented.

Minor comments:

1/ the authors should compare the expression levels of all the Ndc80, Nuf2, and PRC1 mutants introduced via cRNA injections to ensure they are comparable.

Thank you for this comment. We have now performed Western blotting of NDC80-WT/9D/9A (**new Fig. EV4E**), NDC80ΔN-WT/10A, NUF2ΔN-WT/11A (**new Fig. EV6D**), and PRC1-WT/2A/2D mutants (**new Fig. EV7D, H**). Their expression levels were comparable between the mutant and wild-type forms.

2/ could the authors compare the amount of microtubules at BD +2h in all the conditions where bipolarization is delayed? My impression is that there are less microtubules around chromosomes in all these conditions (Fig 1D, 2D, 3C & D). If this is the case, could they check whether it is the nucleation and/or stability of microtubules that is affected in these conditions. This could be an interesting observation to further understand and discuss the mechanism by which timely spindle bipolarization occurs in acentrosomal meiotic spindles. Such an observation could also potentially slightly modify their interpretation as well as their model presented in Fig. 5.

Thank you for this important question. We have quantified the signals of the microtubule marker EGFP-MAP4 in the images shown in Fig. 1C&D and 2D. As shown in **new Fig. EV4A**, microtubule signals were significantly weaker in NDC80-9D+reversine oocytes, compared to control oocytes, at 2 hours after NEBD, consistent with the reviewer's impression. However, such a reduction in microtubule signals was not pronounced at 1 hour after NEBD (**new Fig. EV4A**). Similar observations were made in NDC80ΔN/NUFΔN+reversine oocytes that resulted in severe spindle defects shown in Fig. 2D (**new Fig. EV5G**). These results suggest that reversine-treated oocytes with no stable NDC80-microtubule binding can normally initiate microtubule nucleation but fail to maintain nucleated microtubules, which is associated with severe spindle defects.

In the revised manuscript, we now explicitly state that MPS1 is required for microtubule maintenance in oocytes defective in stable kinetochore-microtubule attachment in the main text and the legend of

Fig. 5. This has greatly improved our discussion of possible mechanisms by which MPS1 promotes spindle bipolarization. Thank you very much. Revised/added sentences are as follows:

“MPS1-inhibited, NDC80-9D-expressing oocytes failed to bipolarize the spindle and exhibited an irregularly shaped spindle throughout meiosis I (Fig. 1D and Movie EV2), suggesting severe spindle bipolarization defects. These oocytes appeared to normally increase nucleated microtubules until 1 hour after nuclear envelope breakdown (NEBD) but failed to accumulate them to full levels by 2 hours after NEBD (Fig. EV4A), suggesting their defects in microtubule maintenance.” (L159–165)

“These observations suggest that MPS1 is required for microtubule maintenance and spindle bipolarization in the absence of stable kinetochore-microtubule attachment during meiosis I in oocytes.” (L175–177)

“...MPS1 inhibition severely impaired the ability of NDC80ΔN-SPC25C and NUF2ΔN-SPC24C to rescue spindle bipolarization, resulting in the formation of an irregularly shaped spindle throughout meiosis I (Fig. 2D). This phenotype was associated with the reduced ability of NDC80ΔN-SPC25C and NUF2ΔN-SPC24C to maintain nucleated microtubules (Fig. EV5G). These results suggest that MPS1 contributes to microtubule maintenance and spindle bipolarization via the C-terminal domains of NDC80-NUF2 at kinetochores...” (L208–214)

“...kinetochores with unstable microtubule attachment employ MPS1 to promote microtubule maintenance and spindle bipolarization via the C-terminal regions of NDC80-NUF2...” (L891–893, Legend of Figure 5)

“...unstably attached kinetochores employ MPS1 activity to create a microenvironment that concentrates microtubule regulators, such as the antiparallel microtubule crosslinker PRC1, via NDC80-NUF2 (Yoshida et al, 2020). This microenvironment facilitates microtubule maintenance and KIF11-mediated bipolar microtubule sorting” (L351–355, Discussion)

EV4A

EV5G

Fig. EV4A. MPS1 is required for microtubule maintenance in the absence of stable kinetochore-microtubule attachments. Live imaging was performed on *Ndc80^{fl} Zp3-Cre* oocytes expressing NDC80-WT/-9D and treated with reversine, using the microtubule marker EGFP-MAP4 (images are shown in Fig. 1C&D). Normalized EGFP-MAP4 intensities are shown (mean \pm SD, n=21, 25, 25, 23 from 3 independent experiments) are shown.

Fig. EV5G. MPS1 is required for microtubule maintenance in the absence of stable kinetochore-microtubule attachments. Live imaging was performed on *Ndc80^{fl} Zp3-Cre* oocytes expressing NDC80ΔN-SPC25C and NUF2ΔN-SPC24C and treated with reversine, using the microtubule marker EGFP-MAP4 (images are shown in Fig. 2D). Normalized EGFP-MAP4 intensities are shown (mean \pm SD, n=26, 26 from 3 independent experiments). The experiment was performed in the presence of proTAME.

We have also quantified the fluorescence signals of mNeonGreen-PRC1 in the images shown in Fig. 3C&D. Consistent with the idea that MPS1 contributes to microtubule maintenance in oocytes deficient in stable kinetochore-microtubule attachment, mNeonGreen-PRC1 signals, which are presumably on microtubules, were significantly reduced by reversine in *Ndc80*-deleted oocytes at 2 hours after NEBD (Fig. 1A for reviewer). We did not observe such a reduction by introducing the 2A mutation to mNeonGreen-PRC1 (Fig. 1B for reviewer). Although these observations might add some ideas to our conclusion, mNeonGreen-PRC1 is not a general marker for microtubules (it preferentially labels antiparallel microtubule bundles), making it difficult to interpret the results. Considering that our new analysis with EGFP-MAP4 mentioned above has better addressed your comment, we have decided not to include these results in the manuscript.

Fig. 1 for reviewer

A. Live imaging was performed on *Ndc80^{ff} Zp3-Cre* oocytes expressing mNeonGreen-PRC1 (green) and H2B-mCherry (chromosome, magenta) and treated with reversine (images are shown in Fig. 3C). Normalized PRC1 intensities are shown (mean \pm SD, n=28, 28 from 3 independent experiments) are shown.

B. As in A, mNeonGreen-PRC1-WT/-2A (T578 and S583 substituted to alanine)-expressing *Ndc80^{ff} Zp3-Cre* oocytes were tested (images are shown in Fig. 3D). Normalized PRC1 intensities are shown (mean \pm SD, n=22, 24 from 3 independent experiments) are shown.

Referee #2:

The accurate segregation of chromosomes depends on spindle biorientation, where each kinetochore pair forms a bipolar attachment to the spindle. In oocytes, which lack centrosomes, kinetochores are initially encircled by randomly oriented microtubules, often resulting in incorrect attachments and potential chromosome segregation errors. Despite this, the factors responsible for initiating spindle biorientation have remained elusive. This study proposes that the kinase MPS1 plays a pivotal role in facilitating spindle bipolarization prior to stabilizing these erroneous attachments, through the regulation of key components such as NDC80, NUF2, and PRC1. In the absence of MPS1, spindle bipolarization is delayed, leading to defective microtubule attachments. The authors introduce a two-step model for spindle assembly in oocytes: In the initial phase, kinetochores with unstable attachments rely on MPS1 to recruit microtubule regulators, aiding in the formation of a bipolar spindle. In the second phase, stable attachments complete spindle assembly and prevent errors. In the first phase, MPS1 is suggested to regulate the proper kinetochore localization of NDC80 and NUF2, phosphorylate their C-terminal regions, and control PRC1 activity, establishing MPS1 as a critical factor in spindle bipolarization. The manuscript is well-constructed, with carefully designed and elegantly executed experiments. This is an exciting study and a good candidate for publication in EMBO. Below are some specific concerns:

Major Points:

In Figure 1b, the spindle is absent in reversine-treated oocytes compared to the control at "timepoint 0". This suggests that MPS1 inhibition by reversine delays spindle assembly. Could this delay explain the postponed bipolarization in reversine-treated oocytes?

Thank you for this comment. We have quantified the signals of the microtubule marker EGFP-MAP4 during the early stages including the timepoint 0 in the image datasets of Fig. 1B, and found no significant delay in the initial onset of microtubule polymerization in reversine-treated oocytes (**new Fig. EV1E**). This means that the images shown in Fig. 1B of the previous manuscript were not very representative – we apologize for this confusion. The revised manuscript now shows the images of a more representative oocyte (**revised Fig. 1B**). We also have added the following statements:

"MPS1 inhibition under the proTAME-treated condition significantly delayed spindle bipolarization (Fig. 1B and Movie EV1), **without a detectable delay in initial microtubule nucleation (Fig. EV1E).**"
(L123–125)

Fig. 1B. MPS1 inhibition delays spindle bipolarization. Live imaging of oocytes with proTAME and reversine. Insets show 3D reconstructed spindles. Temporal changes in the aspect ratio of the spindle (mean \pm SD, $n=14$, 14 oocytes from 2 independent experiments) and morphology classification at 6 hours after nuclear envelope breakdown (NEBD) ($n=26$, 26 from 4 independent experiments) are shown. $**p=0.0034$ by Fisher's exact test for "bipolar" groups. Scale bar, 10 μ m.

Fig. EV1E. Initial microtubule nucleation is not significantly affected by MPS1 inhibition. Live imaging was performed on oocytes treated with proTAME and reversine (images are shown in Fig. 1B). Normalized EGFP-MAP4 intensities are shown (mean \pm SD, $n=26$, 26 oocytes from 4 independent experiments).

In addition, this comment (as well as the comment 2 from reviewer 1) led us to carefully quantify microtubule signals in the oocytes where spindle bipolarization was severely defective. As shown in **new Fig. EV4A**, microtubule signals were significantly weaker in NDC80-9D+reversine oocytes, compared to control oocytes, at 2 hours after NEBD, consistent with the reviewer's impression. However, such a reduction in microtubule signals was not pronounced at 1 hour after NEBD (**new Fig. EV4A**). Similar observations were made in NDC80 Δ N/NUF Δ N+reversine oocytes that resulted in severe spindle defects shown in Fig. 2D (**new Fig. EV5G**). These results suggest that reversine-treated oocytes with no stable NDC80-microtubule binding can normally initiate microtubule nucleation but fail to maintain nucleated microtubules, which is associated with severe spindle defects.

In the revised manuscript, we now explicitly state that MPS1 is required for microtubule maintenance in oocytes defective in stable kinetochore-microtubule attachment in the main text and the legend of Fig. 5. This has greatly improved our discussion of possible mechanisms by which MPS1 promotes spindle bipolarization. Thank you very much.

Revised/added sentences are as follows:

"MPS1-inhibited, NDC80-9D-expressing oocytes failed to bipolarize the spindle and exhibited an irregularly shaped spindle throughout meiosis I (Fig. 1D and Movie EV2), suggesting severe spindle bipolarization defects. These oocytes appeared to normally increase nucleated microtubules until 1 hour after nuclear envelope breakdown (NEBD) but failed to accumulate them to full levels by 2 hours after NEBD (Fig. EV4A), suggesting their defects in microtubule maintenance." (L159–165)

"These observations suggest that MPS1 is required for microtubule maintenance and spindle bipolarization in the absence of stable kinetochore-microtubule attachment during meiosis I in

EV4A

EV5G

Fig. EV4A. MPS1 is required for microtubule maintenance in the absence of stable kinetochore-microtubule attachments. Live imaging was performed on *Ndc80^{fl/fl} Zp3-Cre* oocytes expressing NDC80-WT/-9D and treated with reversine, using the microtubule marker EGFP-MAP4 (images are shown in Fig. 1C&D). Normalized EGFP-MAP4 intensities are shown (mean \pm SD, n=21, 25, 25, 23 from 3 independent experiments) are shown.

Fig. EV5G. MPS1 is required for microtubule maintenance in the absence of stable kinetochore-microtubule attachments. Live imaging was performed on *Ndc80^{fl/fl} Zp3-Cre* oocytes expressing NDC80 Δ N-SPC25C and NUF2 Δ N-SPC24C and treated with reversine, using the microtubule marker EGFP-MAP4 (images are shown in Fig. 2D). Normalized EGFP-MAP4 intensities are shown (mean \pm SD, n=26, 26 from 3 independent experiments). The experiment was performed in the presence of proTAME.

oocytes.” (L175–177)

“...MPS1 inhibition severely impaired the ability of NDC80 Δ N-SPC25C and NUF2 Δ N-SPC24C to rescue spindle bipolarization, resulting in the formation of an irregularly shaped spindle throughout meiosis I (Fig. 2D). This phenotype was associated with the reduced ability of NDC80 Δ N-SPC25C and NUF2 Δ N-SPC24C to maintain nucleated microtubules (Fig. EV5G). These results suggest that MPS1 contributes to microtubule maintenance and spindle bipolarization via the C-terminal domains of NDC80-NUF2 at kinetochores...” (L208–L214)

“...kinetochores with unstable microtubule attachment employ MPS1 to promote microtubule maintenance and spindle bipolarization via the C-terminal regions of NDC80-NUF2...” (L891–893, Legend of Figure 5)

“...unstably attached kinetochores employ MPS1 activity to create a microenvironment that concentrates microtubule regulators, such as the antiparallel microtubule crosslinker PRC1, via NDC80-NUF2 (Yoshida et al, 2020). This microenvironment facilitates microtubule maintenance and KIF11-mediated bipolar microtubule sorting” (L351–355, Discussion)

Is this role in spindle bipolarization specific to MPS1, or could other spindle assembly checkpoint proteins also be involved?

Thank you for this important question. We have performed TRIM-Away-mediated knockdown of the spindle assembly checkpoint protein MAD2. TRIM-Away of MAD2 significantly accelerated anaphase onset (new Fig. EV3A), indicating successful perturbation of the spindle assembly checkpoint. However, unlike reversine-treated oocytes, MAD2-depleted oocytes did not show a detectable delay in spindle bipolarization under proTAME treatment (new Fig. EV3B). These results suggest that the role of MPS1 in spindle bipolarization is independent of the spindle assembly checkpoint.

The following sentences have been added to the manuscript:

“We wondered whether MPS1 promotes spindle bipolarization via the spindle assembly

Fig. EV3. Defective spindle checkpoint does not delay spindle bipolarization

A. MAD2 TRIM-Away accelerates anaphase onset in mouse oocytes. Representative z-projection images of EGFP-MAP4 (spindle, green) and H2B-mCherry (chromosome, magenta) are shown. MAD2 antibody was co-injected with TRIM21 mRNA for MAD2 TRIM-Away (T.A.). Timing of anaphase onset after NEBD are shown (n=18, 21 oocytes from 3 independent experiments). ****p= 0.00000000003 by two-tailed unpaired Mann Whitney test.

B. MAD2 is not required for spindle bipolarization in mouse oocytes. Live imaging of oocytes with proTAME. Temporal changes in the aspect ratio of the spindle (mean \pm SD, n=14, 14 oocytes) and morphology classification at 6 hours after NEBD (n=27, 28 oocytes) are shown. Four independent experiments were performed. n.s., not significant by Fisher's exact test for "bipolar" groups. Scale bars, 10 μ m.

checkpoint. To address this possibility, we knocked down MAD2, a protein essential for the spindle assembly checkpoint (Homer et al, 2005), by TRIM-Away (Clift et al, 2017). MAD2 TRIM-Away significantly accelerated anaphase onset in proTAME-free oocytes (Fig. EV3A), indicating efficient disruption of the spindle assembly checkpoint (Homer et al, 2005). However, in contrast to MPS1 inhibition, MAD2 TRIM-Away did not significantly delay spindle bipolarization in proTAME-treated oocytes (Fig. EV3B). These results suggest that the role of MPS1 in spindle bipolarization is independent of the spindle assembly checkpoint." (L138–145)

The authors propose that MPS1 is crucial for spindle bipolarization. Previous studies have shown that both NDC80 and PRC1 are involved in this process. The authors suggest that MPS1 regulates spindle bipolarization through NDC80 and PRC1. Their results indicate that MPS1 affects the localization and phosphorylation of NDC80 at its C-terminal domain. Although phosphorylation is not required for bipolarization, the localization of the C-terminal domains of NDC80-NUF2 appears to act downstream of MPS1. However, the exact mechanism by which MPS1 regulates this process through NDC80-NUF2 remains unclear. Is it possible that MPS1 recruits PRC1 via the C-terminal domain of NDC80-NUF2?

Yes, it is likely that MPS1 recruits PRC1 via the C-terminal domain of NDC80-NUF2, because

- 1) Expression of the C-terminal domain of NDC80-NUF2 rescues PRC1 recruitment in *Ndc80*-deleted oocytes (Yoshida et al., 2020 Nat Commun, PMID: 32461611; also shown in Fig. EV5F in

this manuscript).

- 2) The C-terminal domain of NDC80-NUF2 interacts with PRC1 in yeast two-hybrid assay (Yoshida et al., 2020 Nat Commun, PMID: 32461611).
- 3) MPS1 activity is required for PRC1 recruitment (Fig. 3A in this manuscript).

We have added the following discussions in the revised manuscript.

“It is likely that MPS1 recruits PRC1 via the C-terminal domains of NDC80-NUF2, as these domains, when tethered to kinetochores, rescue the kinetochore localization defects of PRC1 in Ndc80-deleted oocytes (Yoshida et al, 2020) in an MPS1-dependent manner (Fig. 3A and EV5F). Although a physical interaction of the C-terminal domains of NDC80-NUF2 with PRC1 is suggested by the yeast two-hybrid assay (Yoshida et al, 2020), future work should test this with purified proteins.” (L404–409)

Recently, the same group demonstrated that the NDC80-Nuf2 complex is sufficient to establish spindle biorientation (Asai K et al., 2024) using microbeads coated with NDC80-Nuf2 proteins. These microbeads were able to align at the metaphase plate independently of both the inner and outer kinetochore, as well as the inner centromere. I am wondering whether Mps1 is required to promote the biorientation of NDC80-Nuf2 microbeads?

We reported that NDC80-NUF2-tethered microbeads align at the metaphase plate with cold-unstable microtubule attachment (Asai et al., 2024, PMID: 39298589). We found that reversine did not increase misalignment of NDC80-NUF2-tethered microbeads (**new Fig. EV8A, B**). This result shows that MPS1 activity is dispensable for NDC80-NUF2-tethered microbeads to establish their alignment.

We have incorporated these data and the following texts in the revised manuscript:

“We considered the possibility that MPS1 prevents chromosome misalignment by directly regulating NDC80-NUF2-mediated microtubule attachment. We recently reported that NDC80-NUF2-tethered microbeads efficiently align at the metaphase plate with cold-unstable microtubule attachments (Asai et al, 2024), allowing us to test the requirement of MPS1 for NDC80-NUF2-mediated alignment mechanisms. We found that MPS1 inhibition did not significantly increase the misalignment of NDC80-NUF2-tethered microbeads (Fig. EV8A, B). These results suggest that MPS1 is dispensable for NDC80-NUF2-mediated mechanisms that promote alignment with cold-unstable microtubule attachments.” (L320–327)

Could the authors investigate how MPS1 regulates the localization of both NDC80 and PRC1 at the kinetochore?

To address this question, we need to identify MPS1-mediated phosphorylation sites critical for each of NDC80 and PRC1 localizations. Unfortunately, despite our substantial efforts, the phosphorylation sites that were identified in this study were not required for their localization. The revised manuscript

now clearly states that this important question remains:

“Furthermore, the MPS1-mediated phosphorylation sites on NDC80, NUF2 or PRC1 identified in this study were not required for their kinetochore localization, suggesting that critical phosphorylation sites remain to be identified.” (L409–412)

In Figure S4B, the kinetochore staining of NDC80-T567 is weak but not completely absent in the NDC80 knockout. This raises questions about the specificity of the antibody used. The authors note that phosphorylation is reduced during metaphase I as MPS1 activity declines. How does this dephosphorylation occur, and which phosphatase might be responsible?

Thank you for these comments. First, we agree with the reviewer that the specificity of our NDC80-T567 phospho-antibody is not perfect. We have carefully revised our statements:

“Although reversine treatment or *Ndc80* deletion did not completely abolish the phospho-antibody signals at kinetochores (Fig. 2F and EV6A), substitution of NDC80-T567 with alanine substantially reduced the phospho-antibody signals at kinetochores in oocytes (Fig. EV6B), demonstrating ~~the~~ ~~specificity~~ of that a substantial fraction of the phospho-antibody signals were derived from phosphorylated NDC80-T567. Levels of phosphorylated NDC80-T567 phospho-antibody signals at kinetochores were high during prometaphase and decreased during metaphase (Fig. 2G)...” (L225–231)

Second, a prime candidate for a phosphatase responsible for the gradual dephosphorylation of NDC80-T567 is the PP2A-B56 phosphatase, which is gradually recruited to kinetochores during metaphase I. We can reduce the level of PP2A-B56 recruitment by expressing a phospho-mutant form of the BUBR1, the PP2A-B56 receptor at kinetochores (Yoshida et al., 2015 Dev Cell, PMID: 26028219). We found that the expression of the phospho-mutant form BUBR1-3A significantly increased the level of the NDC80-T567 phospho-antibody signals (new Fig. EV6C). This result is consistent with the idea that the gradual recruitment of PP2A-B56 to kinetochores promotes the gradual dephosphorylation of NDC80-T567 during metaphase I.

We have added the following sentence:

“Overexpression of BUBR1-3A, a phospho-deficient mutant form of BUBR1 that reduces the kinetochore localization of PP2A-B56 phosphatase (Yoshida et al., 2015), significantly increased the kinetochore levels of NDC80-T567 phospho-antibody signals at late metaphase (Fig. EV6C),

consistent with the idea that the gradual dephosphorylation of NDC80-T567 during metaphase is mediated by PP2A-B56.” (L232–237)

The authors show that PRC1 is a potential substrate of MPS1, based on *in vitro* experiments. Specifically, they identify two phosphorylation sites, T578 and S583, that may regulate spindle bipolarization downstream of MPS1. However, it remains to be confirmed if this occurs *in vivo*. Another intriguing question is how these phosphorylations regulate PRC1's function as an antiparallel microtubule cross-linker.

First, we addressed whether we could detect *in vivo* PRC1 phosphorylation by using phosphate affinity polyacrylamide gel electrophoresis (Phos-tag SDS-PAGE) followed by Western blotting. However, it did not detect any MPS1-dependent band shifts of PRC1 (new Fig. EV7F). We still lack direct evidence that T578 and S583 are phosphorylated *in vivo*. This point is clearly stated in the revised manuscript.

“phosphate affinity polyacrylamide gel electrophoresis (Phos-tag SDS-PAGE) followed by Western blotting using prometaphase oocyte extracts did not detect MPS1-dependent band shifts of PRC1 (Fig. EV7F), providing no direct evidence for phosphorylation of these sites *in vivo*.” (L279–282)

To further test whether these two sites could be regulated by phosphorylation *in vivo*, we constructed PRC1-2D, which carries phospho-mimetic aspartic acid substitutions at T578 and S583. PRC1-2D rescued spindle bipolarization defects in *Ndc80*-deleted oocytes (new Fig. EV7G, H), unlike PRC1-2A and similar to PRC1-WT (Fig. 3D), consistent with the idea that these sites are regulated by phosphorylation. The revised manuscript now shows these data with the following statement:

“Nevertheless, PRC1-2D, which carries phospho-mimetic aspartic acid substitutions at these sites, was able to rescue spindle bipolarization defects in *Ndc80*-deleted oocytes (Fig. EV7G, H), unlike PRC1-2A and similar to PRC1-WT (Fig. 3D), consistent with the idea that these sites are regulated by phosphorylation.” (L282–286)

EV7G

Fig. EV7G. PRC1-2D can rescue spindle bipolarization in *Ndc80*-deleted oocytes. Live imaging of *Ndc80*^{fl/fl} *Zp3-Cre* oocytes expressing mNeonGreen-PRC1-WT/2A (T578 and S583 substituted to aspartic acid, green) and H2B-mCherry (chromosome, magenta). Insets show 3D reconstructed images. Temporal changes in the sphericity of the spindle (mean \pm SD, $n=14$, 13 oocytes from 2 independent experiments) and morphology classification at 5 hours after NEBD ($n=24$, 21 oocytes from 3 independent experiments) are shown. n.s., not significant by Fisher's exact test for "bipolar" groups.

Second, thank you for the question on how these phosphorylations could regulate PRC1's function. As the reviewer notes, wild-type PRC1 crosslinks antiparallel microtubules, which are enriched in the middle region of the spindle. If the phospho-mutant form of PRC1 lacks the preferential localization to the middle region of the spindle, it would suggest that phosphorylation is required for PRC1 to facilitate anti-parallel microtubule crosslinking. Conversely, if the mutant form retains the preferential localization to the middle region of the spindle, it raises a possibility that phosphorylation is required for PRC1 to facilitate post-crosslinking processes, such as KIF11-mediated anti-parallel microtubule sliding. We found that PRC1-2A retained its ability to preferentially localize to the middle region of the spindle (**new Fig. EV7E**). Accordingly, we have added this new data with the following statement:

"PRC1-2A preferentially localized to microtubules in the middle region of the spindle (Fig. EV7E), similar to PRC1-WT, suggesting that the mutations did not affect the ability of PRC1 to crosslink anti-parallel microtubules but may have affected its activity in facilitating post-crosslinking processes such as KIF11-mediated anti-parallel microtubule sliding." (L275–279)

EV7E

Fig. EV7E. PRC1-2A preferentially localizes to the middle region of the spindle. Oocytes expressing mNeonGreen-PRC1-WT/2A were fixed. Images of mNeonGreen-PRC1 and Hoechst33342 (DNA) are shown. In each oocyte, intensities of PRC1-WT or -2A along the spindle axis (yellow line) were normalized with their maximum intensity (mean \pm SD, $n=9$, 9 oocytes from 3 independent experiments). PRC1 intensities on the spindle were compared between PRC1-WT and -2A. n.s., not significant by two-tailed unpaired Mann Whitney test.

Minor Points:

The results in Figure S5C would be more convincing with quantification of PRC1-WT and PRC1-8A localization at the kinetochore.

We have quantified the kinetochore signals of PRC1-WT and -8A (revised Fig. EV7C [previously Fig. S5C]). They were comparable.

“Substitution of all 8 candidate phosphorylation sites (PRC1-8A) did not largely affect the ability of PRC1 to localize to kinetochores (Fig. EV7C).” (L262–263)

Repeating the experiments at least three times and increasing the number of quantified oocytes will strengthen the conclusions.

We have now repeated almost all experiments at least three times. The only two exceptions are the experiments for Fig. EV5D and EV8A,B, which we performed twice. We believe that the current dataset is sufficient to support our conclusion, although we would be happy to perform additional experiments for Fig. EV5D and EV8A,B, if requested by the reviewer. The number of experiment replicates is indicated in each figure legend.

Referee #3:

Yoshida et al. addresses an important question of spindle bipolarization relates to correct kinetochore-microtubule attachments in acentrosomal mice oocytes. They provide strong evidence that during prometaphase, when kinetochore-microtubule end-on attachments are not yet established, the MPS1 kinase initiates timely bipolarization of the acentrosomal spindle by promoting NDC80 and PRC1 recruitment to the kinetochore. The Cter of NDC80, but not the MPS1-mediated phosphorylation of the Cter domain, is sufficient for promoting spindle bipolarization. MPS1 also phosphorylates T578 and S583 on PRC1 to promote its spindle bipolarization activity. Interestingly, this study reveals that incorrect but stable kinetochore-microtubule attachments can promote spindle bipolarization, which in turn further stabilizes the kinetochore-microtubule attachments.

Because error correction of incorrect kinetochore-microtubule attachments and the spindle assembly checkpoint surveillance is less stringent in acentrosomal oocytes, this is an important paper that elucidates how MPS1 acts through the NDC80-PRC1 pathway to initiate an early switch to spindle bipolarization to limit the number of attachment errors, thus lowering the initial number of incorrect attachments formed. The manuscript is well written and easy to follow. The findings reveal new mechanistic insight into the timing of spindle bipolarization and justifies how its temporal regulation improves chromosome alignment and protects against aneuploidy in meiosis.

Main points-

1. A reminder of the NDC80-9D and NDC80-9A mutants is required to properly understand the significance of Fig 1C and S3B-C. I cannot find the MDC80-9D mutant in the citations provided (Cheeseman et al., 2006; DeLuca et al., 2006) and any reference for NDC80-9A is missing.

Thank you for pointing this out. We have cited the paper Sundin et al., 2011 MBC (PMID: 21270439), in which NDC80-9D and NDC80-9A are documented. In addition, we have cited our previous paper Courtois et al., 2021 EMBO Rep (PMID: 33655692) where NDC80-9D and -9A are validated in mouse oocytes. Revised statements are:

“NDC80-9D, a phospho-mimetic mutant form deficient in stabilizing kinetochore-microtubule attachment (Cheeseman et al, 2006; DeLuca et al, 2006; Sundin et al, 2011; Courtois et al, 2021)” (L151–153)

“NDC80-9A-expressing oocytes where kinetochore-microtubule attachments are hyperstabilized (Cheeseman et al, 2006; DeLuca et al, 2006; Sundin et al, 2011; Courtois et al, 2021)” (L167–169)

2. Fig 2 claims that Mps1 promotes spindle bipolarization via Cter domains on both NDC80 and NUF2. While all experiments were conducted with both NDC80 and NUF2 mutants, the reduced localization is only assayed for NDC80. The current manuscript does not show any evidence of Mps1 regulating NUF2 localization at the kinetochore. The effects on NUF2 should also be shown in the experiments or the conclusions should be revised to not include NUF2.

Thank you for raising this point. We now show that the level of NUF2 at kinetochores was significantly reduced but remained substantially in reversine-treated oocytes at 1 hour after NEBD (new Fig. EV5C), similar to NDC80 (Fig. EV5A), consistent with the idea that NDC80 and NUF2 behave similarly in the oocyte by forming their heterodimers. The revised manuscript now shows this data with the following statement:

“Similarly, kinetochore NUF2 levels were significantly reduced by MPS1 inhibition just after M-phase entry (Fig. EV5C).” (L185–186)

EV5C

Fig. EV5C. MPS1 inhibition decreases NUF2 localization at kinetochores. Immunostaining of NUF2-HA, ACA (kinetochores), and Hoechst33342 (DNA) in oocytes treated with reversine. Normalized relative intensities of NUF2-HA are shown (median and quartiles, $n=280$, 280 kinetochores from 7, 7 oocytes). Three independent experiments were performed. **** $p < 0.000000001$ by two-tailed unpaired Student's t-test. Scale bar, 10 μm .

3. A more detailed reminder of how KIFC/HSET accelerate spindle bipolarization would be helpful. Fig 4D reports improved chromosome alignment in MPS1-inhibited cells when KIF1C is overexpressed. It is important to clarify if this phenotype is because Kif1C directly regulates chromosome alignment in a MPS1-independent pathway or a result of spindle bipolarization.

Thank you for this suggestion. We have added a more detailed introduction of how KIFC1/HSET accelerates spindle bipolarization.

“KIFC1/HSET is a minus-end-directed microtubule crosslinking motor that is critical for spindle pole focusing (Goshima et al, 2005). When overexpressed in oocytes, it accelerates spindle bipolarization by promoting the sorting of microtubule organizing centers to the spindle poles, depending on its microtubule sliding activity (Bennabi et al, 2018; So et al, 2022).” (L305–309)

Given these established functions of KIFC1/HSET, we favor the idea that the improvement of chromosome alignment by overexpressed KIFC1/HSET overexpression in MPS1-inhibited oocytes is a result of rescued spindle bipolarization.

With these results, we cannot formally exclude the possibility that KIFC1 directly improved chromosome alignment independently of accelerating spindle bipolarization, although we favor the idea that KIFC1-dependent acceleration of spindle bipolarization improved chromosome alignment. We now discuss these points in Discussion:

“in oocytes, a substantial fraction of misaligned chromosomes induced by MPS1 inhibition are likely attributed to delayed spindle bipolarization, as chromosome misalignment and incorrect kinetochore-microtubule attachment in MPS1-inhibited oocytes were largely suppressed by overexpression of KIFC1, which artificially accelerated spindle bipolarization. Although we cannot formally exclude the possibility that KIFC1 directly promotes chromosome alignment, based on the activity of KIFC1 to directly promote microtubule crosslinking and sliding (Goshima et al, 2005; Cai et al, 2009; Bennabi et al, 2018; So et al, 2022), we suggest that KIFC1 suppressed chromosome misalignment by accelerating spindle bipolarization.” (L377–385)

4. In all relevant experiments, the timing when reversine is added to the cells should be included in the text/ figure legend, as this will clarify whether MPS1 activity promotes recruitment of NDC80/PRC1 to the kinetochore or does MPS1 activity prevent removal of NDC80/PRC1 from the kinetochore.

We added reversine immediately after the induction of meiotic resumption (IBMX washout) in almost all the experiments. We have included this information in the main text, Figure legend and Materials and Methods.

“We added reversine to the oocyte culture immediately after inducing meiotic resumption.” (L109–110, Main text)

“Drugs were added to oocyte culture immediately after induction of meiotic resumption.” (L906–907, Figure Legend)

“Unless otherwise indicated, the drugs were added to the oocyte culture immediately after the induction of meiotic resumption.” (L444–445, Materials and Methods)

5. Is Mps1 is only required to initiate the spindle bipolarization or is it also required to maintain the bipolarized spindle? Adding reversine delays bipolarization. Does adding reversine after bipolarization reverse the bipolar spindle to a sphere-shaped spindle and/or weaken the stabilized kinetochore-microtubule attachments?

Thank you for this question. We now show that reversine addition after spindle bipolarization did not reverse the bipolar spindle to a sphere-shaped spindle (**new Fig. EV2H**). Neither a significant increase in incorrect kinetochore-microtubule attachments nor a decrease in correct attachments was observed (**new Fig. EV8C**). We have incorporated these results in the revised manuscript.

“addition of reversine after metaphase spindle establishment did not significantly affect its bipolar shape for the next 2 hours (Fig. EV2H), suggesting that MPS1 activity is not required to maintain spindle bipolarity.” (L132–134)

“Reversine treatment after metaphase spindle establishment did not significantly increase incorrect attachments of kinetochores with cold-stable microtubules nor decrease correct attachments (Fig. EV8C). These results suggest that MPS1 activity after spindle bipolarization is dispensable for suppressing incorrect kinetochore-microtubule attachment.” (L329–333)

EV2H

EV8C

Fig. EV2H. MPS1 is not required for bipolar spindle maintenance. Live imaging of oocytes expressing EGFP-MAP4 (microtubules, green) and H2B-mCherry (chromosomes, magenta). Oocytes were cultured in the presence of proTAME. Reversine was added at 6 hours after NEBD (metaphase I). Temporal changes in the aspect ratio of the spindle after reversine addition (mean \pm SD, $n=15$, 13 oocytes from 3 independent experiments) are shown. Scale bar, 10 μ m.

Fig. EV8C. MPS1 inhibition after metaphase spindle establishment does not increase kinetochore-microtubule attachment errors. Oocytes were cultured in the presence of proTAME. Reversine was added at 6 hours after NEBD (metaphase I). Oocytes were collected 1 hour after the reversine addition, treated briefly with a cold buffer, and then immunostained for stable microtubules (green), kinetochores (magenta), and DNA (blue). Magnified images of end-on monopolar attachments (correct, gray frame) are shown (mean \pm SD, $n=7$, 7 oocytes from 3 independent experiments). n.s., not significant, **** $p=0.001166$ by two-tailed unpaired Mann Whitney test. Scale bar, 10 μ m.

6. PRC1 overexpression rescues the spindle bipolarization in *Ndc80*-deleted oocytes without its kinetochore localization (Yoshida et al., 2020). However, MPS1 phosphorylation of PRC1 is required for PRC1's spindle bipolarization activity. Can the authors speculate on whether PRC1 needs to be at the kinetochore for it to get phosphorylated by MPS1 or do MPS1-PRC1 interact in other parts of the cell?

Our data are consistent with the idea that MPS1 can phosphorylate PRC1 in the cytoplasm because MPS1 inhibition perturbs PRC1-dependent spindle bipolarization in *Ndc80*-deleted oocytes (Fig. 3), where PRC1 does not localize to kinetochores (Yoshida et al., Nat Commun 2020, PMID: 32461611). This does not exclude the possibility that MPS1 phosphorylates PRC1 at kinetochores in normal oocytes. We have discussed these points in the revised manuscript:

“Since MPS1 inhibition perturbs PRC1-dependent spindle bipolarization in *Ndc80*-deleted oocytes (Fig. 3C), where PRC1 does not localize to kinetochores (Yoshida et al, 2020), MPS1 likely can phosphorylate PRC1 in the cytoplasm, although the kinetochore localization of MPS1 likely facilitates the phosphorylation of PRC1, NDC80 and NUF2 at kinetochores in normal oocytes.” (L412–416)

7. Is the MPS1-PRC1 pathway equally important for limiting kinetochore-microtubule attachment errors in metaphase II? A similar analysis as Fig 4B and 4E for meiosis II spindles will be very interesting. Looking at the role of MPS1 in the timing of spindle bipolarization in mitotic cells that have impaired error correction or spindle checkpoint will also reveal if MPS1 acts as a backup pathway for securing chromosome capture by microtubules before anaphase onset in centrosomal cells.

Since spindle bipolarization during meiosis II is inherently fast and does not rely on kinetochores, unlike during meiosis I (Yoshida et al., 2020 Nat Comm, PMID:32461611; Heald et al., 1996 Nature, PMID:868448), we favor the idea that MPS1 is less critical for spindle bipolarization during meiosis II. Nevertheless, it is possible that MPS1 acts as a backup pathway during meiosis II, as suggested by the reviewer. Also, we agree with the reviewer that MPS1's role in timely spindle bipolarization may contribute to correct kinetochore-microtubule attachment as a backup pathway in mitosis. Although we have not investigated mitotic cells as this would be beyond the scope of this study, we have added a discussion of the possible backup role of MPS1 in spindle bipolarization in centrosomal mitotic cells. The revised manuscript includes the following discussion:

“Although this study showed the requirement of MPS1 for timely spindle bipolarization during meiosis I in oocytes, it remains unclear whether MPS1 plays a similar role in meiosis II and somatic mitosis. In contrast to meiosis I, bipolar spindle assembly in meiosis II does not rely on kinetochores but is largely mediated by chromosome-dependent pathways (Yoshida et al., 2020 Nat Comm; Heald et al., 1996), suggesting the possibility that the kinetochore MPS1-dependent pathway may be less critical for spindle bipolarization in meiosis II. In somatic mitosis with centriole-containing centrosomes, centrosomes play an important role in bipolar spindle assembly. We suggest that the MPS1-mediated timely initiation of spindle bipolarization is particularly critical for meiosis I in acentrosomal oocytes. It is still possible that the MPS1-dependent kinetochore pathway acts as a back-up mechanism for bipolar spindle assembly in meiosis II and somatic mitosis.” (L386–396)

Dr. Tomoya S Kitajima
RIKEN Center for Biosystems Dynamics Research
Laboratory for Chromosome Segregation
2-2-3 Minatojima-minamimachi, Chuo-ku
Kobe
650-0047
Japan

22nd Apr 2025

Re: EMBOJ-2024-118908R
MPS1 promotes timely spindle bipolarization to prevent kinetochore-microtubule attachment errors in oocytes

Dear Tomoya,

Thank you for submitting your revised manuscript to The EMBO Journal. Two of the original referees have now assessed it once again (see comments below), and both of them are overall satisfied with the revisions. Referee 2 still has two specific queries, which I would invite you to respond to in a response letter and, if applicable, in the manuscript text. Furthermore, there are still a few editorial issues to be addressed prior to formal acceptance:

- Please change the headers for the Material/Methods sections simply into 'Methods'.
- As we are switching from a free-text author contribution statement towards a more formal statement based on Contributor Role Taxonomy (CRediT) terms, please remove the present Author Contribution section and instead specify each author's contribution(s) directly in the Author Information page of our submission system during upload of the final manuscript. See <https://casrai.org/credit/> for more information.
- Finally, please provide suggestions for a short 'blurb' text prefacing and summing up the study in two sentences (max. 250 characters), followed by 3-5 one-sentence 'bullet points' with brief factual statements of key results of the paper; they will form the basis of an editor-written 'Synopsis' accompanying the online version of the article. Please also upload a synopsis image, which can be used as a "visual title" for the synopsis section of your paper. The image should be in PNG or JPG format with the modest dimensions of EXACTLY 550 pixels wide and 300-600 pixels high (this could be simply a slightly condensed/simplified version of Figure 5).

I am therefore returning the manuscript to you for a final round of revision, solely to allow you to make these modifications and upload the revised files. Once we will have received them, we should be ready to swiftly proceed with formal acceptance and production of the manuscript.

With kind regards,

Hartmut

- 2) Each figure legend must specify
- size of the scale bars that are mandatory for all micrograph panels
 - the statistical test used to generate error bars and P-values
 - the type error bars (e.g., S.E.M., S.D.)

- the number (n) and nature (biological or technical replicate) of independent experiments underlying each data point
- Figures may not include error bars for experiments with $n < 3$; scatter plots showing individual data points should be used instead.

9) To facilitate reproducibility and cross-laboratory adoption of methodologies, please structure the Materials & Methods section as outlined in our guide to authors, including a completed Reagents and Tools Table that can be downloaded from our author guidelines as well (<https://www.embopress.org/page/journal/14602075/authorguide#structuredmethods>).

10) Digital image enhancement is acceptable practice, as long as it accurately represents the original data and conforms to community standards. If a figure has been subjected to significant electronic manipulation, this must be clearly noted in the figure legend and/or the 'Materials and Methods' section. The editors reserve the right to request original versions of figures and the original images that were used to assemble the figure. Finally, we generally encourage uploading of numerical as well as gel/blot image source data; for details see: embopress.org/page/journal/14602075/authorguide#sourcedata

In the interest of ensuring the conceptual advance provided by the work, we recommend submitting a revision within 3 months (21st Jul 2025). Please discuss the revision progress ahead of this time with the editor if you require more time to complete the revisions. Use the link below to submit your revision:

Link Not Available

Referee #2:

This is an important article that provides insight into how spindle bipolarization occurs during meiosis I in mouse oocytes. The authors have made impressive efforts with additional experiments to strengthen the manuscript. While they have properly addressed most of the previous concerns and I remain enthusiastic about this interesting study, I have two points that require further clarification:

1. Could the authors explain how Mps1 inhibition reduces the kinetochore localization of NDC80 at 1 hour after NEBD (Fig. EV5B), but not at 2 hours after NEBD (Fig. EV7A)?

2. In lines 212-214, the authors suggest that Mps1 promotes spindle bipolarization via the C-terminal domain of NDC80-NUF2.

However, targeting this domain to the kinetochore does not rescue spindle bipolarization following Mps1 inhibition. Should the authors conclude that Mps1 regulates spindle bipolarization not merely through the localization of NDC80-NUF2, but rather via activation of the NDC80-NUF2 C-terminus, which may be required to recruit other factors (e.g., PRC1) essential for spindle bipolarization?

Referee #3:

The authors provided satisfactory responses to all the points raised. The revisions were very thorough and well-executed. The revised conclusions and discussion are now carefully worded and accurately reflect the experimental data. Thus, the manuscript has been significantly improved and provides important insight into fundamental molecular mechanisms of chromosome-microtubule dynamics and should be accepted for publication.

Point-by-point response to reviewer's comments

EMBOJ-2024-118908

MPS1 promotes timely spindle bipolarization to prevent kinetochore-microtubule attachment errors in oocytes

Correspondence to: tomoya.kitajima@riken.jp

Plain letters indicate reviewers' comments.

Blue letters indicate our responses.

"Red letters enclosed in quotation marks" indicate texts newly added to the revised manuscript.

We thank the reviewers for their positive and constructive suggestions comments.

Referee #2:

This is an important article that provides insight into how spindle bipolarization occurs during meiosis I in mouse oocytes. The authors have made impressive efforts with additional experiments to strengthen the manuscript. While they have properly addressed most of the previous concerns and I remain enthusiastic about this interesting study, I have two points that require further clarification:

1. Could the authors explain how Mps1 inhibition reduces the kinetochore localization of NDC80 at 1 hour after NEBD (Fig. EV5B), but not at 2 hours after NEBD (Fig. EV7A)?

Thank you for your question. We speculate that MPS1 is responsible for ensuring the quick kinetochore localization of NDC80 upon NEBD but is not essential for its recruitment mechanism. We have clarified this point by revising a sentence in **Discussion**:

"MPS1 kinase activity promotes spindle bipolarization likely through multiple pathways in oocytes. First, MPS1 ensures rapid and proper localization of NDC80 to kinetochores within the short time window of one hour after NEBD. Second..." (L394–396)

2. In lines 212-214, the authors suggest that Mps1 promotes spindle bipolarization via the C-terminal domain of NDC80-NUF2. However, targeting this domain to the kinetochore does not rescue spindle bipolarization following Mps1 inhibition. Should the authors conclude that Mps1 regulates spindle bipolarization not merely through the localization of NDC80-NUF2, but rather via activation of the NDC80-NUF2 C-terminus, which may be required to recruit other factors (e.g., PRC1) essential for spindle bipolarization?

Thank you for requesting clarification. Yes, it is exactly our conclusion. We believe that this conclusion is clearly documented in **Discussion**:

"MPS1 kinase activity promotes spindle bipolarization likely through multiple pathways in oocytes. First, MPS1 ensures rapid and proper localization of NDC80 to kinetochores within the short time window of one hour after NEBD. Second, MPS1 regulates the C-terminal regions of NDC80 and NUF2. Our data show that MPS1 directly phosphorylates these regions in oocytes, although it is unclear whether direct phosphorylation contributes to timely spindle bipolarization. Third, MPS1 regulates PRC1, an antiparallel microtubule crosslinker that promotes spindle bipolarization (Yoshida et al, 2020; Bieling et al, 2010; Subramanian et al, 2010). MPS1

promotes kinetochore localization and spindle bipolarizing activity of PRC1.” (L394–402)

Referee #2:

The authors provided satisfactory responses to all the points raised. The revisions were very thorough and well-executed. The revised conclusions and discussion are now carefully worded and accurately reflect the experimental data. Thus, the manuscript has been significantly improved and provides important insight into fundamental molecular mechanisms of chromosome-microtubule dynamics and should be accepted for publication.

Thank you for acknowledging our efforts. We appreciate your helpful comments.

Dr. Tomoya S Kitajima
RIKEN Center for Biosystems Dynamics Research
Laboratory for Chromosome Segregation
2-2-3 Minatojima-minamimachi, Chuo-ku
Kobe
650-0047
Japan

25th Apr 2025

Re: EMBOJ-2024-118908R1
MPS1 promotes timely spindle bipolarization to prevent kinetochore-microtubule attachment errors in oocytes

Dear Dr. Kitajima,

Thank you for submitting your final revised manuscript for our consideration. I am pleased to inform you that we have now accepted it for publication in The EMBO Journal.

Yours sincerely,

Hartmut Vodermaier
